# Learning to Receive Help:
# Intervention-Aware Concept Embedding Models

**Mateo Espinosa Zarlenga**
University of Cambridge
me466@cam.ac.uk

**Katherine M. Collins**
University of Cambridge
kmc61@cam.ac.uk

**Krishnamurthy (Dj) Dvijotham**
Google DeepMind
dvij@google.com

**Adrian Weller**
University of Cambridge
Alan Turing Institute
aw665@cam.ac.uk

**Zohreh Shams**
University of Cambridge
zs315@cam.ac.uk

**Mateja Jamnik**
University of Cambridge
mateja.jamnik@cl.cam.ac.uk

## Abstract

Concept Bottleneck Models (CBMs) tackle the opacity of neural architectures by constructing and explaining their predictions using a set of high-level concepts. A special property of these models is that they permit *concept interventions*, wherein users can correct mispredicted concepts and thus improve the model's performance. Recent work, however, has shown that intervention efficacy can be highly dependent on the order in which concepts are intervened on and on the model's architecture and training hyperparameters. We argue that this is rooted in a CBM's lack of train-time incentives for the model to be appropriately receptive to concept interventions. To address this, we propose Intervention-aware Concept Embedding models (IntCEMs), a novel CBM-based architecture and training paradigm that improves a model's receptiveness to test-time interventions. Our model learns a concept intervention policy in an end-to-end fashion from where it can sample meaningful intervention trajectories at train-time. This conditions IntCEMs to effectively select and receive concept interventions when deployed at test-time. Our experiments show that IntCEMs significantly outperform state-of-the-art concept-interpretable models when provided with test-time concept interventions, demonstrating the effectiveness of our approach.

## 1 Introduction

It is important to know how to ask for help, but also important to know how to *receive* help. Knowing how to react to feedback positively allows for bias correction [1] and efficient learning [2] while being instrumental for mass collaborations [3]. Nevertheless, although the uptake of feedback is ubiquitous in real-world decision-making, the same cannot be said about modern artificial intelligence (AI) systems, where deployment tends to be in isolation from experts whose feedback could be queried.

Progress in this aspect has recently come from Explainable AI, where interpretable Deep Neural Networks (DNNs) that can benefit from expert feedback at test-time have been proposed [4–7]. In particular, Concept Bottleneck Models (CBMs) [4], a family of interpretable DNNs, enable expert-model interactions by generating, as part of their inference process, explanations for their predictions using high-level "concepts". Such explanations allow human experts to better understand a CBM's prediction based on interpretable units of information (e.g., input is a "cat" because it has "paws" and "whiskers") rather than low-level features (e.g., input pixels). This design enables humans to provide feedback to the CBM at test-time via *concept interventions* [4] (Figure 1), a process specific to CBM-like architectures in which an expert analyses a CBM's inferred concepts

37th Conference on Neural Information Processing Systems (NeurIPS 2023).

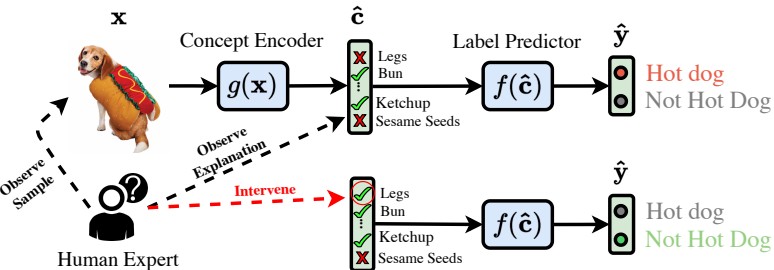

Figure 1: When intervening on a CBM, a human expert analyses the predicted concepts and corrects mispredicted values (e.g., the mispredicted concept "legs"), allowing the CBM to update its prediction.

and corrects mispredictions. Concept interventions allow a CBM to update its final prediction after incorporating new expert feedback, and have been shown to lead to significant performance boosts [4, 5, 8, 9]. Nevertheless, and in contrast to these results, recent works have found that concept interventions can potentially *increase* a CBM's test-time error depending on its choice of architecture [10] and its set of train-time annotations [5]. This discrepancy suggests that a CBM's receptiveness to interventions is neither guaranteed nor fully understood.

In this paper, we posit that the sensitivity of a CBM's receptiveness to interventions is an artefact of its training objective function. Specifically, we argue that CBMs *lack an explicit incentive to be receptive to interventions* as they are neither exposed to interventions at train-time nor optimised to perform well under interventions. To address this, we propose a novel intervention-aware objective function and architecture based on Concept Embedding Models (CEMs) [5], a generalisation of CBMs that represent concepts as high-dimensional embeddings. Our architecture, which we call *Intervention-aware Concept Embedding Model* (IntCEM), has two key distinctive features. First, it learns an intervention policy end-to-end which imposes a prior over which concepts an expert could be queried to intervene on next to maximise intervention receptiveness. Second, its loss function includes an explicit regulariser that penalises IntCEM for mispredicting the task label after a series of train-time interventions sampled from its learned policy. We highlight the following contributions:

- We introduce IntCEM, the first concept-interpretable neural architecture that learns not only to explain its predictions using concepts but also learns an intervention policy dictating which concepts an expert should intervene on next to maximise test performance.
- We show that IntCEM significantly outperforms all baselines in the presence of interventions while maintaining competitiveness when deployed without any interventions.
- We demonstrate that, by rewriting concept interventions as a differentiable operation, IntCEM learns an efficient dynamic intervention policy that selects performance-boosting concepts to intervene on at test-time.

## 2   Background and Previous Work

**Concept Bottleneck Models**   Methods in Concept-based Explainable AI [11–17] advocate for explaining DNNs by constructing explanations for their predictions using high-level concepts captured within their latent space. Within these methods, Concept Bottleneck Models (CBMs) [4] provide a framework for developing concept-based interpretable DNNs.

Given a training set $\mathcal{D} := \big\{(\mathbf{x}^{(i)}, \mathbf{c}^{(i)}, y^{(i)})\big\}_{i=1}^N$, where each sample $\mathbf{x} \in \mathbb{R}^n$ (e.g., an image) is annotated with a task label $y \in \{1, \cdots, L\}$ (e.g., "cat" or "dog") and $k$ binary concepts $\mathbf{c} \in \{0,1\}^k$ (e.g., "has whiskers"), a CBM is a pair of functions $(g, f)$ whose composition $f(g(\mathbf{x}))$ predicts a probability over output classes given $\mathbf{x}$. The first function $g : \mathbb{R}^n \to [0,1]^k$, called the *concept encoder*, learns a mapping between features $\mathbf{x}$ and concept activations $\hat{\mathbf{c}} = g(\mathbf{x}) \in [0,1]^k$, where ideally $\hat{c}_i$ is close to $1$ when concept $c_i$ is active in $\mathbf{x}$, and $0$ otherwise. The second function $f : [0,1]^k \to [0,1]^L$, called the *label predictor*, learns a mapping from the predicted concepts $\hat{\mathbf{c}}$ to a distribution $\hat{\mathbf{y}} = f(\hat{\mathbf{c}})$ over $L$ task classes. When implemented as DNNs, $g$'s and $f$'s parameters can be learnt (i) *jointly*, by minimising a combination of the concept predictive loss $\mathcal{L}_{\text{concept}}$ and the task cross-entropy loss $\mathcal{L}_{\text{task}}$, (ii) *sequentially*, by first training $g$ to minimise $\mathcal{L}_{\text{concept}}$ and then training

$f$ to minimise $\mathcal{L}_{\text{task}}$ from $g$'s outputs, or (iii) *independently*, where $g$ and $f$ are independently trained to minimise their respective losses. Because at prediction-time $f$ has access only to the "bottleneck" of concepts activations $\hat{\mathbf{c}}$, the composition $(f \circ g)$ yields a concept-interpretable model where $\hat{c}_i$ can be interpreted as the probability $\hat{p}_i$ that concept $c_i$ is active in $\mathbf{x}$.

**Concept Embedding Models**   Because $f$ operates only on the concept scores predicted by $g$, if the set of training concepts is not predictive of the downstream task, then a CBM will be forced to choose between being highly accurate at predicting concepts *or* at predicting task labels. Concept Embedding Models (CEMs) [5] are a generalisation of CBMs that address this trade-off using high-dimensional concept representations in their bottlenecks. This design allows information of concepts not provided during training to flow into the label predictor via two $m$-dimensional vector representations (i.e., embeddings) for each training concept $c_i$: $\mathbf{c}_i^+ \in \mathbb{R}^m$, an embedding representing $c_i$ when it is active (i.e., $c_i = 1$), and $\mathbf{c}_i^- \in \mathbb{R}^m$, an embedding representing $c_i$ when it is inactive (i.e., $c_i = 0$).

When processing sample $\mathbf{x}$, for each training concept $c_i$ a CEM constructs $\mathbf{c}_i^+$ and $\mathbf{c}_i^-$ by feeding $\mathbf{x}$ into two learnable models $\phi_i^+, \phi_i^- : \mathbb{R}^n \to \mathbb{R}^m$ implemented as DNNs with a shared preprocessing module. These embeddings are then passed to a learnable scoring model $s : \mathbb{R}^{2m} \to [0, 1]$, shared across all concepts, that predicts the probability $\hat{p}_i = s([\hat{\mathbf{c}}_i^+, \hat{\mathbf{c}}_i^-])$ of concept $c_i$ being active. With these probabilities, one can define a CEM's concept encoder $g(\mathbf{x}) = \hat{\mathbf{c}} := [\hat{\mathbf{c}}_1, \cdots, \hat{\mathbf{c}}_k] \in \mathbb{R}^{km}$ by mixing the positive and negative embeddings of each concept to generate a final concept embedding $\hat{\mathbf{c}}_i := \hat{p}_i \hat{\mathbf{c}}_i^+ + (1 - \hat{p}_i)\hat{\mathbf{c}}_i^-$. Finally, a CEM can predict a label $\hat{\mathbf{y}}$ for sample $\mathbf{x}$ by passing $\hat{\mathbf{c}}$ (i.e., its "bottleneck") to a learnable label predictor $f(\hat{\mathbf{c}})$ whose output can be explained with the concept probabilities $\hat{\mathbf{p}} := [\hat{p}_1, \cdots, \hat{p}_k]^T$. This generalisation of CBMs has been empirically shown to outperform CBMs especially when the set of concept annotations is incomplete [5].

**Concept Interventions**   A key property of both CBMs and CEMs is that they allow *concept interventions*, whereby experts can correct mispredicted concept probabilities $\hat{\mathbf{p}}$ at test-time. By updating the predicted probability of concept $c_i$ (i.e., $\hat{p}_i$) so that it matches the ground truth concept label (i.e., setting $\hat{p}_i := c_i$), these models can update their predicted bottleneck $\hat{\mathbf{c}}$ and propagate that change into their label predictor $g(\hat{\mathbf{x}})$, leading to a potential update in the model's output prediction.

Let $\mu \in \{0, 1\}^k$ be a mask with $\mu_i = 1$ if we will intervene on concept $c_i$, and $\mu_i = 0$ otherwise. Here, assume that we are given the ground truth values $\tilde{\mathbf{c}}$ of all the concepts we will intervene on. For notational simplicity, and without loss of generality, we can extend $\tilde{\mathbf{c}}$ to be in $[0, 1]^k$ by setting $\tilde{c}_i = c_i$ if $\mu_i = 1$, and $\tilde{c}_i = 0.5$ otherwise (where $0.5$ expresses full uncertainty but, as it will become clear next, the specific value is of no importance). Thus, this vector contains the expert-provided ground truth concept labels for concepts we are intervening on, while it assigns an arbitrary value (e.g., $0.5$) to all other concepts. We define an intervention as a process where, for all concepts $c_i$, the activation(s) $\hat{\mathbf{c}}_i$ corresponding to concept $c_i$ in a bottleneck $\hat{\mathbf{c}}$ are updated as follows:

$$\hat{\mathbf{c}}_i := (\mu_i \tilde{c}_i + (1 - \mu_i)\hat{p}_i)\hat{\mathbf{c}}_i^+ + \big(1 - (\mu_i \tilde{c}_i + (1 - \mu_i)\hat{p}_i)\big)\hat{\mathbf{c}}_i^- \tag{1}$$

where $\hat{\mathbf{c}}_i^+$ and $\hat{\mathbf{c}}_i^-$ are $c_i$'s positive and negative concept embeddings (for CBMs $\hat{\mathbf{c}}_i^+ = [1]$ and $\hat{\mathbf{c}}_i^- = [0]$). This update forces the mixing coefficient between the positive and negative concept embeddings to be the ground truth concept value $c_i$ when we intervene on concept $c_i$ (i.e., $\mu_i = 1$), while maintaining it as $\hat{p}_i$ if we are not intervening on concept $c_i$. The latter fact holds as $(\mu_i \tilde{c}_i + (1 - \mu_i)\hat{p}_i) = \hat{p}_i$ when $\mu_i = 0$, regardless of $\tilde{c}_i$. Because the predicted positive and negative concept embeddings, as well as the predicted concept probabilities, are all functions of $\mathbf{x}$, for simplicity we use $\tilde{g}(\mathbf{x}, \mu, \tilde{\mathbf{c}})$ to represent the updated bottleneck of CBM $(g, f)$ when intervening with mask $\mu$ and concept values $\tilde{\mathbf{c}}$.

We note that although we call the operation defined in Equation 1 a "concept intervention" to follow the term used in the concept learning literature, it should be distinguished from interventions in causal models [18–20]. Instead, concept interventions in this context are connected, yet not identical, to previous work in active feature acquisition in which one has access to an expert at test-time to request the ground truth value of a small number of input features [21–23].

**Intervention Policies**   In practice, it may not be sensible to ask a human to intervene on *every* concept [24]; instead, we may want a preferential ordering of which concepts to query first (e.g., some concepts may be highly informative of the output task, yet hard for the model to predict). "Intervention policies" have been developed to produce a sensible ordering for concept interventions [8, 25, 9].

These orderings can be produced independently of a particular instance (i.e., a *static* policy), or selected on a per-instance basis (a *dynamic* policy). The latter takes as an input a mask of previously intervened concepts $\mu$ and a set of predicted concept probabilities $\hat{\mathbf{p}}$, and determines which concept should be requested next from an expert, where the retrieved concept label is assumed to be "ground truth". Two such policies are *Cooperative Prediction* (CooP) [9] and *Expected Change in Target Prediction* (ECTP) [8], which prioritise concept $c_i$ if intervening on $c_i$ next leads to the largest change in the probability of the currently predicted class in expectation (with the expectation taken over the distribution given by $p(c_i = 1) = \hat{p}_i$). These policies, however, may become costly as the number of concepts increases and share the limitation that, by maximising the expected change in probability in the predicted class, they do not guarantee that the probability mass shifts towards the correct class.

# 3 Intervention-Aware Concept Embedding Models

Although there has been significant work on understanding how CBMs react to interventions [5, 6, 9, 25], the fact that these models are trained without any expert intervention – yet are expected to be highly receptive to expert feedback at test-time – is often overlooked. To the best of our knowledge, the only exception comes from Espinosa Zarlenga et al. [5], wherein the authors introduced *RandInt*, a procedure that *randomly* intervenes on the model during train-time. However, RandInt assumes that a model should be equally penalised for mispredicting $y$ regardless of how many interventions have been performed. This leads to CEMs lacking the incentive to perform better when given more feedback. Moreover, RandInt assumes that all concepts are equally likely to be intervened on. This inadequately predisposes a model to assume that concepts will be randomly intervened on at test-time.

To address these limitations, we propose Intervention-Aware Concept Embedding Models (IntCEMs), a new CEM-based architecture and training framework designed for inducing *receptiveness to interventions*. IntCEMs are composed of two core components: 1) an end-to-end learnable concept intervention policy $\psi(\hat{\mathbf{c}}, \mu)$, and 2) a novel objective function that penalises IntCEM if it mispredicts its output label after following an intervention trajectory sampled from $\psi$ (with a heavier penalty if the trajectory is longer). By learning the intervention policy $\psi$ in conjunction with the concept encoder and label predictor, IntCEM can *simulate* dynamic intervention steps at train-time while it learns a policy which enables it to "ask" for help about specific concepts at test-time.

## 3.1 Architecture Description and Inference Procedure

An IntCEM is a tuple of parametric functions $(g, f, \psi)$ where: (i) $g$ is a concept encoder, mapping features $\mathbf{x}$ to a bottleneck $\hat{\mathbf{c}}$, (ii) $f$ is a label predictor mapping $\hat{\mathbf{c}}$ to a distribution over $L$ classes, and (iii) $\psi : \mathbb{R}^{km} \times \{0, 1\}^k \rightarrow [0, 1]^k$ is a concept intervention policy mapping $\hat{\mathbf{c}}$ and a mask of previously intervened concepts $\mu$ to a probability distribution over which concepts to intervene on next. IntCEM's concept encoder $g$ works by (1) passing an input $\mathbf{x}$ to a learnable "backbone" $\zeta(\mathbf{x}) := \left(\{(\hat{\mathbf{c}}_i^+, \hat{\mathbf{c}}_i^-)\}_{i=1}^k, \hat{\mathbf{p}}\right)$, which predicts a pair of embeddings $(\hat{\mathbf{c}}_i^+, \hat{\mathbf{c}}_i^-)$ and a concept probability $\hat{p}_i$ for all concepts, and (2) constructs a bottleneck $\hat{\mathbf{c}}$ by concatenating all the mixed vectors $\{\hat{p}_1 \hat{\mathbf{c}}_1^+ + (1 - \hat{p}_1)\hat{\mathbf{c}}_1^-, \cdots, \hat{p}_k \hat{\mathbf{c}}_k^+ + (1 - \hat{p}_k)\hat{\mathbf{c}}_k^-\}$. While multiple instantiations are possible for $\zeta$, in this work we parameterise it using the same backbone as in a CEM. This implies that $\zeta$ is formed by concept embedding generating models $\{(\phi_i^+, \phi_i^-)\}_{i=1}^k$ and scoring model $s$. This yields a model which, at inference, is almost identical to a CEM, except that IntCEM also outputs a *probability distribution* $\psi(\hat{\mathbf{c}}, \mu)$ placing a high density on concepts that may yield significant intervention boosts. At train-time, these models differ significantly, as we discuss next.

## 3.2 IntCEM's Training Procedure

The crux of IntCEM's training procedure, shown in Figure 2, lies in inducing receptiveness to test-time interventions by exposing our model to dynamic train-time interventions. Specifically, at the beginning of each training step, we sample an initial mask of intervened concepts $\mu^{(0)}$ from a prior distribution $p(\mu)$ as well as the number $T \in \{1, \cdots, k\}$ of interventions to perform on top of $\mu^{(0)}$ from prior $T \sim p(T)$. Next, we generate a *trajectory* of $T$ new interventions $\{\eta^{(t)} \in \{0, 1\}^k \sim p(\eta ; \omega^{(t)})\}_{t=1}^T$ by sampling the $t^{\text{th}}$ intervention $\eta^{(t)} \sim p(\eta ; \omega^{(t)})$ (represented as a one-hot encoding) from a categorical distribution with parameters $\omega^{(t)}$. A key component of this work is that parameters $\omega^{(t)}$ are predicted using IntCEM's policy $\psi\big(\hat{\mathbf{c}}^{(t-1)}, \mu^{(t-1)}\big)$ evaluated on the

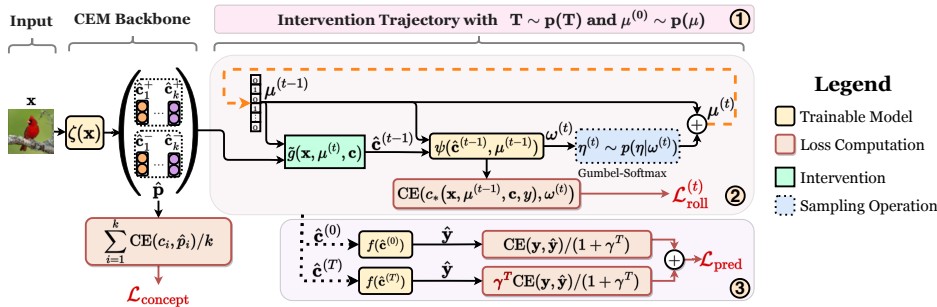

Figure 2: Given concept embeddings and probabilities $\zeta(\mathbf{x}) = \big(\{(\hat{\mathbf{c}}_i^+, \hat{\mathbf{c}}_i^-)\}_{i=1}^k, \hat{\mathbf{p}}\big)$, an IntCEM training step (1) samples an intervention mask $\mu^{(0)} \sim p(\mu)$ and horizon $T \sim p(T)$, (2) generates an intervention trajectory $\{(\mu^{(t-1)}, \eta^{(t)})\}_{t=1}^T$ from the learnable intervention policy $\psi$, (3) and predicts the task label at the start *and* end of the trajectory. Our loss incentivises (i) good initial concept predictions ($\mathcal{L}_{\text{concept}}$), (ii) performance-boosting intervention trajectories ($\mathcal{L}_{\text{roll}}$), (iii) and low task loss before and after interventions ($\mathcal{L}_{\text{pred}}$), with a **heavier penalty** $\gamma^T > 1$ for mispredicting at the end of the trajectory. In this figure, dashed orange arrows indicate recursive steps in our sampling process.

previous bottleneck $\hat{\mathbf{c}}^{(t-1)} := \tilde{g}(\mathbf{x}, \mu^{(t-1)}, \mathbf{c})$ and intervention mask $\mu^{(t-1)}$, where we recursively define $\mu^{(t)}$ as $\mu^{(t)} := \mu^{(t-1)} + \eta^{(t)}$ for $t > 0$. This allows IntCEM to calibrate its train-time trajectory of interventions to reinforce supervision on the embeddings corresponding to currently mispredicted concepts, leading to IntCEM learning more robust representations of inherently harder concepts.

We design our objective function such that it incentivises learning performance-boosting intervention trajectories *and* achieving high task performance as more interventions are provided:

$$\mathcal{L}(\mathbf{x}, \mathbf{c}, y, \mathcal{T}) := \lambda_{\text{roll}} \mathcal{L}_{\text{roll}}(\mathbf{x}, \mathbf{c}, y, \mathcal{T}) + \mathcal{L}_{\text{pred}}(\mathbf{x}, \mathbf{c}, y, \mu^{(0)}, \mu^{(T)}) + \lambda_{\text{concept}} \mathcal{L}_{\text{concept}}(\mathbf{c}, \hat{\mathbf{p}})$$

where $\mathcal{T} := \{(\mu^{(t-1)}, \eta^{(t)})\}_{t=1}^T$ is the intervention trajectory while $\lambda_{\text{roll}}$ and $\lambda_{\text{concept}}$ are user-defined hyperparameters. $\lambda_{\texttt{roll}}$ encodes a user's tradeoff for number of interventions versus task performance gains (e.g., high $\lambda_{\texttt{roll}}$ encourages rapid task performance gains from few interventions, whereas low $\lambda_{\texttt{roll}}$ is more suitable in settings where many expert queries can be made). $\lambda_{\text{concept}}$ expresses a user's preference between accurate explanations and accurate task predictions before any interventions.

**Rollout Loss ($\mathcal{L}_{\text{roll}}$)**   The purpose of $\mathcal{L}_{\text{roll}}$ is to learn an intervention policy that prioritises the selection of concepts for which the associated concept embeddings fails to represent the ground truth concept labels, i.e., select concepts which benefit from human intervention. We achieve this through a train-time "behavioural cloning" [26, 27] approach where $\psi$ is trained to learn to *predict the action taken by an optimal greedy policy* at each step. For this, we take advantage of the presence of ground truth concepts and task labels during training and incentivise $\psi$ to mimic "Skyline", an oracle optimal policy proposed by Chauhan et al. [9]. Given a bottleneck $\hat{\mathbf{c}}$ and ground-truth labels $(y, \mathbf{c})$, the Skyline policy selects $c_*(\mathbf{x}, \mu, \mathbf{c}, y) := \arg\max_{1 \le i \le k} f\big(\tilde{g}(\mathbf{x}, \mu \vee \mathbb{1}_i, \mathbf{c})\big)_y$ as its next concept, where $f(\cdot)_y$ is the probability of class $y$ predicted by $f$ and $\mu \vee \mathbb{1}_i$ represents the action of adding an intervention on concept $i$ to $\mu$. Intuitively, $c_*$ corresponds to the concept whose ground-truth label and intervention would yield the highest probability over the ground-truth class $y$. We use such a demonstration to provide feedback to $\psi$ through the following loss:

$$\mathcal{L}_{\text{roll}}(\mathbf{x}, \mathbf{c}, y, \mathcal{T}) := \frac{1}{T} \sum_{t=1}^T \text{CE}(c_*(\mathbf{x}, \mu^{(t-1)}, \mathbf{c}, y), \psi(\tilde{g}(\mathbf{x}, \mu^{(t-1)}, \mathbf{c}), \mu^{(t-1)}))$$

where $\text{CE}(\mathbf{p}, \hat{\mathbf{p}})$ is the cross-entropy loss between ground truth distribution $\mathbf{p}$ and predicted distribution $\hat{\mathbf{p}}$, penalising $\psi$ for not selecting the same concept as Skyline throughout its trajectory.

**Task Prediction Loss ($\mathcal{L}_{\text{pred}}$)**   We penalise IntCEM for mispredicting $y$, both before and after our intervention trajectory, by imposing a higher penalty when it mispredicts $y$ at the end of the trajectory:

$$\mathcal{L}_{\text{pred}}(\mathbf{x}, \mathbf{c}, y, \mu^{(0)}, \mu^{(T)}) := \frac{\text{CE}\big(y, f(\tilde{g}(\mathbf{x}, \mu^{(0)}, \mathbf{c}))\big) + \gamma^T \text{CE}\big(y, f(\tilde{g}(\mathbf{x}, \mu^{(T)}, \mathbf{c}))\big)}{1 + \gamma^T}$$

Here $\gamma \in [1, \infty)$ is a scaling factor that penalises IntCEM more heavily for mispredicting $y$ at the end of the trajectory, by a factor of $\gamma^T > 1$, than for mispredicting $y$ at the start of the trajectory. In practice, we observe that a value in $\gamma \in [1.1, 1.5]$ works well; we use $\gamma = 1.1$ in our experiments unless specified otherwise. For an ablation study showing that IntCEM's intervention performance surpasses that of existing methods for a wide array of scaling factors $\gamma$, see Appendix A.6.

A key realisation is that expressing interventions as in Equation (1) yields an operator which is *differentiable with respect to the concept intervention mask* $\mu$. This trick allows us to backpropagate gradients from $\mathcal{L}_{\text{pred}}$ into $\psi$ when the sampling operation $\eta^{(t)} \sim p(\eta | \psi(\hat{\mathbf{c}}^{(t-1)}, \mu^{(t-1)}))$ is differentiable with respect to its parameters. In this work, this is achieved by relaxing our trajectory sampling using a differentiable Gumbel-Softmax [28] categorical sampler (see Appendix A.2 for details).

**Concept Loss ($\mathcal{L}_{\text{concept}}$)**  The last term in our loss incentivises accurate concept explanations for IntCEM's predictions using the cross-entropy loss averaged across all training concepts:

$$\mathcal{L}_{\text{concept}}(\mathbf{c}, \hat{\mathbf{p}}) := \frac{1}{k} \sum_{i=1}^{k} \text{CE}(c_i, \hat{p}_i) = \frac{1}{k} \sum_{i=1}^{k} \big( - c_i \log \hat{p}_i - (1 - c_i) \log (1 - \hat{p}_i) \big)$$

**Putting everything together ($\mathcal{L}$)**  We learn IntCEM's parameters $\theta$ by minimising the following:

$$\theta^* = \arg\min_{\theta} \mathbb{E}_{(\mathbf{x}, \mathbf{c}, y) \sim \mathcal{D}} \Big[ \mathbb{E}_{\mu \sim p(\mu), \, T \sim p(T)} \big[ \mathcal{L}(\mathbf{x}, \mathbf{c}, y, \mathcal{T}(\mathbf{x}, \mathbf{c}, \mu^{(0)}, T)) \big] \Big] \qquad (2)$$

The outer expectation can be optimised via stochastic gradient descent, while the inner expectation can be estimated using Monte Carlo samples from the user-selected priors $(p(\mu), p(T))$. For the sake of simplicity, we estimate the inner expectation using *a single Monte Carlo sample per training step* for both the initial mask $\mu^{(0)}$ and the trajectory horizon $T$. In our experiments, we opt to use a Bernoulli prior for $p(\mu)$, where each concept is selected with probability $p_{\text{int}} = 0.25$, and a discrete uniform prior $\text{Unif}(\{1, \cdots, T_{\max}\})$ for $p(T)$, where $T_{\max}$ is annealed within $[2, 6]$. Although domain-specific knowledge can be incorporated into these priors, we leave this for future work and show in Appendix A.6.3 that IntCEMs are receptive to interventions as we vary $p_{\text{int}}$ and $T_{\max}$.

## 4 Experiments

In this section, we evaluate IntCEMs by exploring the following research questions:

- **Unintervened Performance (Q1)**: In the absence of interventions, how does IntCEM's task and concept predictive performance compare to CEMs's and CBM's? Does IntCEM's updated loss function have detrimental effects on its uninterverned performance?
- **Intervention Performance (Q2A)**: Are IntCEMs more receptive to test-time interventions than state-of-the-art CBM variants?
- **Effects of Policies during Deployment (Q2B)**: What is the impact on IntCEM's performance when employing different intervention policies at test-time?
- **Benefits of End-to-End Learning of $\psi$ (Q3)**: Is it beneficial to learn our intervention policy $\psi$ at train-time?

**Datasets and Tasks**  We consider five vision tasks: (1) `MNIST-Add`, a task inspired by the UMNIST dataset [29] where one is provided with 12 MNIST [30] images containing digits in $\{0, \cdots, 9\}$, as well as their digit labels as concept annotations, and needs to predict if their sum is at least half of the maximum attainable, (2) `MNIST-Add-Incomp`, a concept-incomplete [17] extension of the `MNIST-Add` task where only 8/12 operands are provided as concept annotations, (3) `CUB` [31], a bird classification task whose set of concepts cover 112 bird features, e.g., "wing shape", selected by Koh et al. [4], (4) `CUB-Incomp`, a concept-incomplete extension of `CUB` where we provide only 25% of all concept groups, and (5) `CelebA`, a task on the Celebrity Attributes Dataset [32] whose task and concept labels follow those selected in [5]. For more details refer to Appendix A.3.

**Models**  We evaluate IntCEMs against CEMs with the exact same architecture, averaging all metrics of interest over five different random initialisations. We also include CBMs that are trained jointly (*Joint CBM*), sequentially (*Seq. CBM*) and independently (*Ind. CBM*) as part of our evaluation to

Table 1: Task predictive performance (accuracy or AUC in %) and mean concept AUC (%) across all tasks and baselines. Values within one standard deviation from the highest baseline are emphasised. CelebA's task accuracies are relatively low across all baselines given its inherent difficulty (256 highly imbalanced classes). Nevertheless, we highlight these results match previous observations [5].

| | Dataset | IntCEM($\lambda_{roll} = 5$) | IntCEM($\lambda_{roll} = 1$) | IntCEM($\lambda_{roll} = 0.1$) | CEM | Joint CBM-Sigmoid | Joint CBM-Logit | Ind. CBM | Seq. CBM |
|---|---|---|---|---|---|---|---|---|---|
| **Task** | MNIST-Add | 90.74 ± 1.66 | **92.07 ± 0.20** | **92.05 ± 0.23** | 90.13 ± 0.41 | 79.25 ± 2.93 | 86.09 ± 0.60 | 87.66 ± 0.62 | 83.20 ± 1.08 |
| | MNIST-Add-Incomp | **89.41 ± 0.16** | **89.43 ± 0.39** | **89.63 ± 0.13** | 86.61 ± 0.75 | 76.61 ± 1.13 | 84.03 ± 0.87 | 86.49 ± 0.67 | 85.23 ± 0.78 |
| | CUB | 76.20 ± 0.98 | 78.03 ± 0.58 | 77.79 ± 0.19 | **79.00 ± 0.61** | 74.95 ± 0.56 | **78.16 ± 0.24** | 60.25 ± 3.20 | 53.41 ± 4.10 |
| | CUB-Incomp | 70.39 ± 1.11 | 74.70 ± 0.22 | **74.66 ± 0.50** | **75.52 ± 0.37** | 59.73 ± 4.06 | 68.42 ± 7.91 | 44.82 ± 0.33 | 40.00 ± 1.59 |
| | CelebA | **38.09 ± 0.26** | 32.89 ± 0.86 | 30.66 ± 0.41 | 31.04 ± 0.96 | 24.15 ± 0.43 | 23.87 ± 1.26 | 24.25 ± 0.26 | 24.53 ± 0.3 |
| **Concept** | MNIST-Add | 80.81 ± 3.28 | 85.26 ± 0.22 | 85.03 ± 0.27 | 85.51 ± 0.26 | **89.73 ± 0.15** | **89.21 ± 0.71** | 82.29 ± 0.66 | 82.29 ± 0.66 |
| | MNIST-Add-Incomp | 87.07 ± 1.08 | 87.13 ± 1.98 | 87.36 ± 1.44 | 86.98 ± 0.01 | **91.15 ± 0.45** | **90.88 ± 0.25** | 87.58 ± 0.37 | 87.58 ± 0.37 |
| | CUB | 89.63 ± 0.42 | 92.51 ± 0.08 | 93.39 ± 0.22 | **94.46 ± 0.04** | 93.76 ± 0.16 | 93.68 ± 0.09 | 89.85 ± 0.78 | 89.85 ± 0.78 |
| | CUB-Incomp | 92.70 ± 0.19 | **94.42 ± 0.13** | **94.50 ± 0.15** | **94.65 ± 0.10** | 93.58 ± 0.09 | 92.87 ± 2.01 | 93.65 ± 0.25 | 93.65 ± 0.25 |
| | CelebA | **88.25 ± 0.24** | 86.46 ± 0.48 | 85.35 ± 0.07 | 85.28 ± 0.18 | 81.14 ± 0.75 | 82.85 ± 0.44 | 82.85 ± 0.21 | 82.85 ± 0.21 |

be able to compare IntCEM's policy against state-of-art intervention policies developed for CBMs. Since previous work shows that interventions in joint CBMs are sensitive to its bottleneck activation function [4, 10], we include joint CBMs with a sigmoidal (*Joint CBM-Sigmoid*) and logit-based (*Joint CBM-Logit*) bottleneck. We note that we do not include other noteworthy concept-based models such as leakage-free CBMs [6], GlanceNets [33], Self-explaining Neural Networks (SENNs) [34], Concept Whitening (CW) [35], or Concept Model Extraction (CME) [16] as these either lack a well-defined mechanism to be intervened on (e.g., as in SENNs and CW) or represent more constrained versions of other baselines such as CEMs (e.g., leakage-free CBMs, GlanceNets, and CME).

Throughout our experiments, we strive for a fair comparison by using the same architectures for the concept encoder $g$ and label predictor $f$ of all baselines. Unless specified otherwise, for each task, we tune IntCEM's $\lambda_{roll}$ hyperparameter by varying it over $\{5, 1, 0.1\}$ and selecting the model with the highest area under the curve defined by plotting the validation task accuracy vs number of interventions. For the CUB-based tasks and CelebA, we use the same values of $\lambda_{concept}$ selected in [5]. Otherwise, we select this value from $\{10, 5, 1\}$ using CEM's validation error. Following [5], we use embeddings with $m = 16$ activations and a value of $p_{int} = 0.25$ for CEM's RandInt. For further details on model and training hyperparameters, see Appendix A.4.

## 4.1 Task and Concept Performance in the Absence of Interventions (Q1)

First, we examine how IntCEM's loss function affects its *unintervened test performance* compared to existing baselines as we vary $\lambda_{roll}$. We summarise our results in Table 1, where, for all tasks and baselines, we show the downstream task predictive test accuracy, or the AUC when the task is binary (i.e., MNIST-based tasks), *and* the concept performance. We measure the latter via the mean AUC of predicting ground-truth test concept $c_i$ from its corresponding predicted concept probability $\hat{p}_i$.

**In the absence of concept interventions, IntCEM is as accurate as state-of-the-art baselines.** Evaluating test task accuracy reveals that IntCEMs are more accurate (e.g., by ∼7% in CelebA) or just as accurate as other baselines across most tasks. Only in CUB do we observe a small drop in performance (less than ∼1%) for the best-performing IntCEM variant. Notice that three of the five tasks we use, namely MNIST-Add-Incomp, CUB-Incomp, and CelebA, lack a complete set of training concept annotations. Therefore, our results suggest that IntCEM maintains a high task performance even with concept incompleteness – a highly desired property for real-world deployment. Finally, the performance improvements of IntCEMs over CEMs in MNIST, MNIST-Incomp, and CelebA suggest that our loss function may lead to more informative concept representations that enable more accurate label predictors to be trained.

**IntCEM's concept performance is competitive with respect to traditional CEMs.** The bottom half of Table 1 shows that IntCEM's mean concept AUC, particularly for low values of $\lambda_{roll}$, is competitive against that of CEMs. The only exception is in the MNIST-based task, where Joint CBMs outperform CEMs and IntCEMs. Nevertheless, this drop is overshadowed by the poor task accuracy obtained by joint CBMs in these two tasks, highlighting that although CBMs may have better concept predictive accuracy than CEMs, their task performance is generally bounded by that of a CEM. These results demonstrate that IntCEM's explanations are as accurate as those in CEMs without sacrificing task accuracy, suggesting our proposed objective loss does not significantly affect IntCEM's performance in the absence of interventions. Finally, we observe a trade-off between incentivising the IntCEM to be highly accurate after a small number of interventions (i.e., high $\lambda_{roll}$)

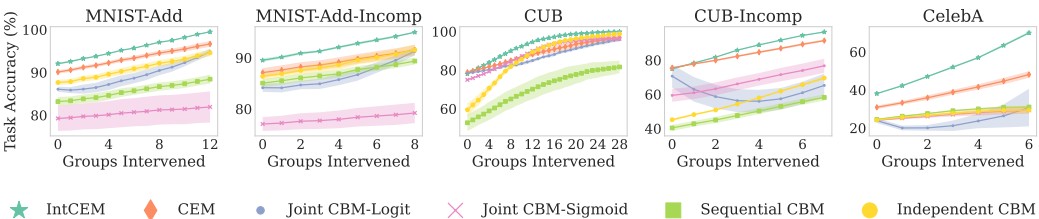

Figure 3: Task accuracy of all baseline models after receiving a varying number of *randomly selected* interventions. For our binary MNIST-based tasks, we show task AUC rather than accuracy. Here and elsewhere, we show the means and standard deviations (can be insignificant) over five random seeds.

and incentivising the IntCEM to generate more accurate concept explanations (i.e., low $\lambda_{\text{roll}}$), which can be calibrated by setting $\lambda_{\text{roll}}$ after considering how the IntCEM will be deployed.

## 4.2 Intervention Performance (Q2)

Given that a core appeal of CBM-like models is their receptiveness to interventions, we now explore how test-time interventions affect IntCEM's task performance compared to other models. We first study whether IntCEM's training procedure preconditions it to be more receptive to test-time interventions by evaluating the task accuracy of all baselines after receiving random concept interventions. Then, we investigate whether IntCEM's competitive advantage holds when exposing our baselines to state-of-the-art intervention policies. Below, we report our findings by showing each model's task accuracy as we intervene, following [4], on an increasing number of groups of mutually exclusive concepts (e.g., "white wing" and "black wing").

**IntCEM is significantly more receptive to test-time interventions.** Figure 3, shows that IntCEM's task accuracy after receiving concept interventions is significantly better than that of competing methods across all tasks, illuminating the benefits of IntCEM preconditioning for receptiveness. We observe this for both concept-complete tasks (i.e., CUB and MNIST-Add) as well as concept-incomplete tasks (i.e., rest). In particular, we see significant gains in CUB and CelebA; IntCEMs attain large performance improvements with only a handful of random interventions (∼10% with ∼25% of concepts intervened) and surpass all baselines.

**IntCEM's performance when intervened on with a randomly selected set of concepts can be better than the theoretically-proven optimal intervention performance achievable by CEMs alone.** We explore IntCEM's receptiveness to different test-time intervention policies by computing its test performance while intervening following: (1) a *Random* intervention policy, where the next concept is selected, uniformly at random, from the set of unknown concepts, (2) the *Uncertainty of Concept Prediction* (UCP) [8] policy, where the next concept is selected by choosing the concept $c_i$ whose predicted probability $\hat{p}_i$ has the highest uncertainty (measured by $1/|\hat{p}_i - 0.5|$), (3) the *Cooperative Policy (CoP)* [9], where we select the concept that maximises a linear combination of its predicted uncertainty (akin to UCP) and the expected change in the predicted label's probability when intervening on that concept, (4) *Concept Validation Accuracy* (CVA) static policy, where concepts are selected using a fixed order starting with those whose validation errors are the highest as done by Koh et al. [4], (5) the *Concept Validation Improvement* (CVI) [9] static policy, where concepts are selected using a fixed order starting with those concepts that, on average, lead to the biggest improvements in validation accuracy when intervened on, and finally (6) an oracle *Skyline* policy, which selects $c_*$ on every step, indicating an upper bound for all greedy policies. For details on each policy baseline, including how their hyperparameters are selected, see Appendix A.7.

We demonstrate in Figure 4 that IntCEM consistently outperforms CEMs regardless of the test-time intervention policy used, with CoP and UCP consistently yielding superior results. Further, we uncover that intervening on IntCEM with a Random policy at test-time *outperforms the theoretical best performance attainable by a CEM (Skyline)* under many interventions, while its best-performing policy outperforms CEM's Skyline after very few interventions. These results suggest that IntCEM's train-time conditioning not only results in IntCEMs being more receptive to test-time interventions

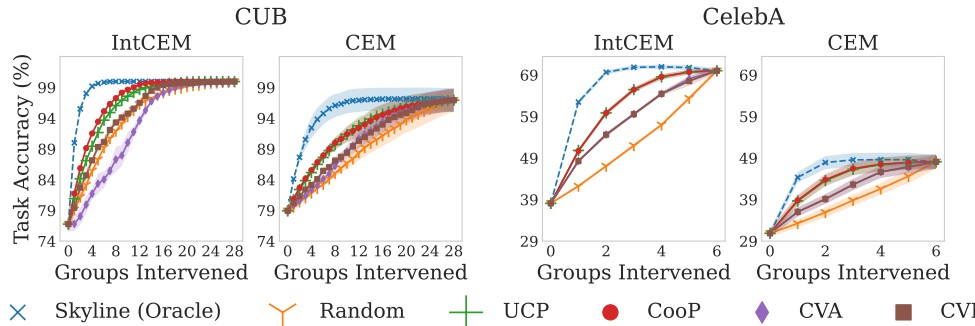

Figure 4: Task accuracy of IntCEMs and CEMs on `CUB` and `CelebA` when intervening with different test-time policies. We show similar improvements of IntCEMs over CBMs in Appendix A.7.

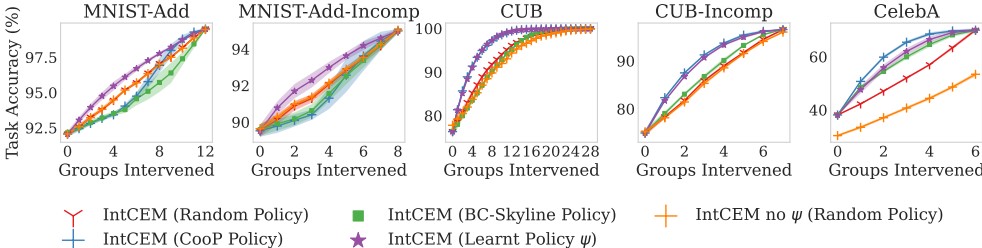

Figure 5: Task performance when intervening on IntCEMs following test-time policies $\psi$, *CooP*, *Random*, and *BC-Skyline*. Our baseline "IntCEM no $\psi$" is an IntCEM whose test *and* train interventions are sampled from a Random policy rather than from $\psi$ (i.e., a policy is not learnt in this baseline).

but may lead to significant changes in an IntCEM's theoretical-optimal intervention performance. We include analyses on our remaining datasets, revealing similar trends, in Appendix A.7.

### 4.3 Studying IntCEM's Intervention Policy (Q3)

Finally, we explore IntCEM's intervention policy by evaluating test-time interventions predicted by $\psi$. To understand whether there is a benefit of learning $\psi$ end-to-end (i.e., in conjunction with the rest of model training), rather than learning it post-hoc *after* training an IntCEM, we compare it against a Behavioural Cloning (BC) [26] policy "*BC-Skyline*" trained on demonstrations of Skyline applied to a trained IntCEM. Furthermore, to explore how $\psi$'s trajectories at *train-time* help improve IntCEM's receptiveness to test-time interventions, we study an IntCEM trained by sampling trajectories uniformly at random ("IntCEM no $\psi$"). Our results in Figure 5 suggest that: (1) IntCEM's learnt policy $\psi$ leads to test-time interventions that are as good or better than CooP's, yet avoid CooP's computational cost (*up to ~2x faster* as seen in Appendix A.5), (2) learning $\psi$ end-to-end during training yields a better policy than one learnt through BC *after* the IntCEM is trained, as seen in the BC policy's lower performance; and (3) using a learnable intervention policy at train-time results in a significant boost of test-time performance. This suggests that part of IntCEM's receptiveness to interventions, even when using a random test-time policy, can be partially attributed to learning and using $\psi$ during training. We show some qualitative examples of rollouts of $\psi$ in the `CUB` dataset in Figure 6.

## 5 Discussion and Conclusion

**Concept Leakage Within IntCEMs**    Previous work [36, 6, 10] has shown that CBMs are prone to encoding unnecessary information in their learnt concept representations. This phenomenon, called *concept leakage*, may lead to less interpretable concept representations [36] and detrimental concept interventions in CBMs [10]. Given that interventions in IntCEMs involve swapping a concept's predicted embeddings rather than overwriting an activation in the bottleneck, such interventions are distinct from those in CBMs as they enable leaked information to be exploited *after* an intervention

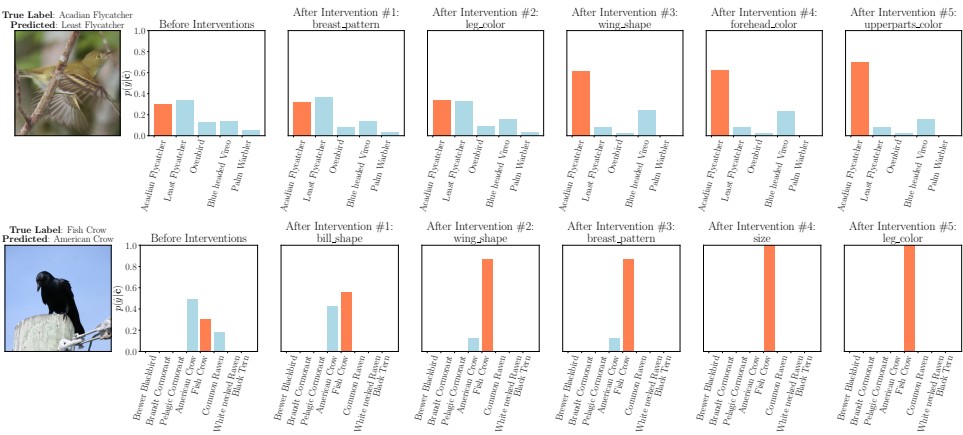

Figure 6: Examples of interventions on an IntCEM following its policy $\psi$ in CUB. We show the task label distribution $p(\hat{y}|\hat{\mathbf{c}})$ for the most likely classes after each intervention. We highlight the correct label's probability using orange bars and show the selected concept by $\psi$ above each panel.

is performed. This means our loss function may incentivise IntCEM to exploit this mechanism to improve a model's receptiveness to interventions. In Appendix A.10 we explore this hypothesis and find that we can detect more leakage in IntCEM's concept representations than in those learnt by CBMs and CEMs. This suggests that, contrary to common assumptions, leakage may be a healthy byproduct of more expressive concept representations in models that accommodate such expressivity. Nevertheless, we believe further work is needed to understand the consequences of this leakage.

**Limitations and Future Work**    To the best of our knowledge, we are the first to frame learning to receive concept interventions as a joint optimisation problem where we simultaneously learn concepts, downstream labels, and policies. IntCEMs offer a principled way to prepare concept-based models for interventions at test-time, without sacrificing performance in the absence of interventions. Here, we focused on CEMs as our base model. However, our method can be extended to traditional CBMs (see Appendix A.8), and future work may consider extending it to more recent concept-based models such as post-hoc CBMs [37], label-free CBMs [38], and probabilistic CBMs [39].

Furthermore, we note some limitations. First, IntCEM requires hyperparameter tuning for $\lambda_{\mathrm{roll}}$ and $\lambda_{\mathrm{concept}}$. While our ablations in Appendix A.6 suggest that an IntCEM's performance gains extend across multiple values of $\lambda_{\mathrm{roll}}$, users still need to tune such parameters for maximising IntCEM's utility. These challenges are exacerbated by the computational costs of training IntCEMs due to their train-time trajectory sampling (see Appendix A.5). However, such overhead gets amortised over time given $\psi$'s efficiency over competing policies. Further, we recognise our training procedure renders IntCEMs more receptive to *any* form of intervention, including adversarial interactions (see Appendix A.9). This raises a potential societal concern when deciding how our proposed model is deployed. Future work may explore such issues by incorporating error-correcting mechanisms or by considering intervention-time human uncertainty [29].

Finally, our evaluation was limited to two real-world datasets and one synthetic dataset, all of which have medium-to-small training set sizes. Therefore, future work may explore how to apply our proposed architecture to larger real-world datasets and may explore how to best deploy IntCEM's policy in practice via large user studies.

**Conclusion**    A core feature of concept-based models is that they permit experts to interpret predictions in terms of high-level concepts and *intervene* on mispredicted concepts hoping to improve task performance. Counter-intuitively, such models are rarely trained with interventions in mind. In this work, we introduce a novel concept-interpretable architecture and training paradigm – Intervention-Aware Concept Embedding Models (IntCEMs) – designed explicitly for intervention receptiveness. IntCEMs simulate interventions during train-time, continually preparing the model for interventions they may receive when deployed, while learning an efficient intervention policy. Given the cost of querying experts for help, our work addresses the lack of a mechanism to leverage help when received, and demonstrates the value of studying models which, by design, know how to utilise help.

## Acknowledgments and Disclosure of Funding

The authors would like to thank Naveen Raman, Andrei Margeloiu, and the NeurIPS 2023 reviewers for their insightful and thorough comments on earlier versions of this manuscript. MEZ acknowledges support from the Gates Cambridge Trust via a Gates Cambridge Scholarship. KMC acknowledges support from the Marshall Commission and the Cambridge Trust. AW acknowledges support from a Turing AI Fellowship under grant EP/V025279/1, The Alan Turing Institute, and the Leverhulme Trust via CFI. GM is funded by the Research Foundation-Flanders (FWO-Vlaanderen, GA No 1239422N). MJ is supported by the EPSRC grant EP/T019603/1.

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

# A Appendix

## A.1 Software and Hardware Used

**Software** For this work, we evaluate all baselines using the MIT-licensed implementations of both CBMs and CEMs made public in [5]. Our implementation of IntCEM is built on top of that repository using PyTorch 1.12 [40], an open-source deep learning library with a BSD license. We incorporated CooP into this library by basing ourselves on the code released under an MIT license by Chauhan et al. [9] as part of their original publication; their code was written in TensorFlow, and as such, we converted it to PyTorch. All numerical plots and graphics have been generated using Matplotlib 3.5, a Python-based plotting library with a BSD license, while all conceptual figures were generated using draw.io, a free drawing software distributed under an Apache 2.0 license. Furthermore, when building our experiment tasks, we use TensorFlow's distribution [41] of MNIST [30] and CelebA [32] and use the processing scripts made public by Koh et al. [4] to generate our concept annotated CUB-tasks. All of our code, including configs and scripts to recreate results shown in this paper, has been released as part of CEM's official public repository found at https://github.com/mateoespinosa/cem.

**Resources** All our non-ablation experiments were run in a shared GPU cluster with four Nvidia Titan Xp GPUs and 40 Intel(R) Xeon(R) E5-2630 v4 CPUs (at 2.20GHz) with 125GB of RAM. In contrast, all of our ablation experiments were run on a separate GPU cluster with 4x Nvidia A100-SXM-80GB GPUs, where each GPU is allocated 32 CPUs. We estimate that to complete all of our experiments, including prototyping and initial explorations, we required approximately 350 to 370 GPU hours.

## A.2 Additional Details on the Gumbel-Softmax Sampler

As introduced in Section 3, expressing interventions using Equation (1) allows us to differentiate the output embedding for concept $c_i$ with respect to the intervention mask $\mu$ *after an intervention has been performed*. Therefore, if the output of the sampler $p(\eta \mid \psi(\hat{\mathbf{c}}^{(t-1)}, \mu^{(t-1)}))$ used to generate intervention $\eta$ is differentiable with respect to its parameters $\psi(\hat{\mathbf{c}}^{(t-1)}, \mu^{(t-1)})$, then we can use the gradients from $\nabla\text{CE}\big(y, f(\tilde{g}(\mathbf{x}, \mu^{(T)}, \mathbf{c}))\big)$, one of the terms in $\mathcal{L}_{\text{pred}}$, to update the parameters of $\psi$ through standard back-propagation. This is because the intervention mask used at the end of the training trajectory is given by $\mu^{(T)} = \mu^{(0)} + \sum_{t=1}^{T} \eta^{(t)}$, allowing the gradient of $\tilde{g}(\mathbf{x}, \mu^{(T)}, \mathbf{c})$ to be back-propagated into the generating functions of each of the interventions $\{\eta^{(t)}\}_{t=1}^{T}$. Therefore, when parameterising our learnable policy $\psi$ as a DNN with weights $\theta_\psi$, this policy can better update $\theta_\psi$ during training so that its interventions decrease both the prediction loss $\mathcal{L}_{\text{pred}}$ at the end of any intervention trajectory as well as its mean rollout loss $\mathcal{L}_{\text{roll}}$.

In practice, we construct a differentiable categorical sampler $p(\eta \mid \psi(\hat{\mathbf{c}}^{(t-1)}, \mu^{(t-1)}))$ using a Gumbel-Softmax distribution [28]: a differentiable and continuous relaxation of the categorical distribution. To use this distribution, we begin by interpreting $\psi(\hat{\mathbf{c}}, \mu) = \omega \in (-\infty, 0)^k$ as the log-probability that each concept will be selected for the next intervention. Then, we proceed by sampling a noise vector $\mathbf{h} \in \mathbb{R}^k$ from a Gumbel distribution such that each dimension $h_i$ of this vector is independently sampled from $h_i \sim \text{Gumbel}(0, 1)$. This noise allows us to introduce non-determinism in our system, akin to the use of Gaussian noise in a Variational Autoencoder's reparameterisation trick [42]. Finally, the crux of the Gumbel-Softmax trick is to use this noise to shift the log-probabilities predicted by $\psi$ and generating a sharp continuous representation of the sample through a softmax activation function:

$$\texttt{GumbelSoftmax}(\omega; \tau) = \text{Softmax}((\omega + \mathbf{h})/\tau)$$

Here, $\tau \in [0, \infty)$ is a *temperature* hyperparameter indicating the sharpness of the output continuous representation of the sample (smaller values leading to samples that are closer to a one-hot representation of a vector) and $\text{Softmax}(\omega)$ is the softmax function whose $i$-th output is given by

$$\text{Softmax}(\omega)_i = \frac{e^{\omega_i}}{\sum_{j=1}^{k} e^{\omega_j}}$$

In this process, one can easily sample the Gumbel noise variables $h_i$ from $\text{Gumbel}(0, 1)$ by first sampling $u_i \sim \text{Uniform}(0, 1)$ and then computing $h_i = -\log(-\log(u_i))$. In this work, we always use a temperature of $\tau = 1$ and make the output of the Gumbel Softmax discrete (i.e., into a one-hot

representation) using the Straight-Through estimator proposed by Jang et al. [28] which still enables continuous gradients to be back-propagated. Note, this procedure permits one to accidentally sample a previously intervened concept from $p(\eta|\psi(\hat{\mathbf{c}}^{(t-1)}, \mu^{(t-1)}))$ (as there may be a non-zero probability assigned to a concept $c_i$ where $\mu_i^{(t-1)} = 1$); to guard against this possibility, after sampling an intervention $\eta^{(t)}$ from the Gumbel Softmax distribution, we clamp all values of the new intervention mask $\mu^{(t)} = \mu^{(t-1)} + \eta^{(t)}$ to be between 0 and 1 for all $t$.

## A.3 Dataset Details

In this Appendix, we describe the datasets and tasks used in Section 4. A summary of each task's properties can be found in Table A.1 and a detailed discussion of each task can be found below.

Table A.1: Details of all tasks used in this paper. For each task, we show the number of testing samples and training samples. Furthermore, we include the shape of each input, the number of concepts provided (as well as how many groups of mutually exclusive concepts there are), and the number of output task labels. In all experiments, we subsample 20% of the training set to make our validation set.

| Task | Training Samples | Testing Samples | Input Size | Concepts ($k$) | Groups | Output Labels |
|---|---|---|---|---|---|---|
| MNIST-Add | 12,000 | 10,000 | [12, 28, 28] | 72 | 12 | 1 |
| MNIST-Add-Incomp | 12,000 | 10,000 | [12, 28, 28] | 54 | 8 | 1 |
| CUB | 5,994 | 5,794 | [3, 299, 299] | 112 | 28 | 200 |
| CUB-Incomp | 5,994 | 5,794 | [3, 299, 299] | 22 | 7 | 200 |
| CelebA | 13,507 | 3,376 | [3, 64, 64] | 6 | 6 | 256 |

**MNIST-Add**  MNIST-Add is a visual arithmetic task based on the MNIST [30] dataset. Each sample $\mathbf{x} \in [0, 1]^{12 \times 28 \times 28}$ in this task is formed by 12 digit "operands", where the $i$-th channel $\mathbf{x}_{i,:,:}$ of $\mathbf{x}$ contains a grayscale zero-to-nine digit representing the $i$-th operand. We construct this dataset such that some operands are inherently harder to predict. To do so, we (i) constrain each of the first four operands to be at most 2, (ii) constrain each of the four middle operands to at most 4, and (iii) constrain each of the last four operands to be at most 9.

Each sample $\mathbf{x}$ in this task is annotated with a binary label $y$ corresponding to whether the sum of the digits in all 12 operands is at least half the maximum attainable (i.e., at least 30). We construct concept annotations $\mathbf{c} \in \{0, 1\}^{72}$ for $\mathbf{x}$ by providing the one-hot encoding representations of each operand in $\mathbf{x}$ as an annotation. Because the first four operands can be at most 2, and the next 4 and 8 operands can be at most 4 and 9, respectively, this results in a total of $4 \times 3 + 4 \times 5 + 4 \times 10 = 72$ concepts organised into groups of 12 mutually exclusive concepts. To construct MNIST-Add's training set, we generate 12,000 training samples by randomly selecting all 12 operands of each sample from MNIST's training set. Similarly, to construct MNIST-Add's test set, we generate 10,000 test samples by randomly selecting all 12 operands of each sample from MNIST's *test* set.

**MNIST-Add-Incomp**  MNIST-Add-Incomp is a visual arithmetic task that modifies MNIST-Add so that its set of concept annotations is incomplete with respect to the task label. This is done by randomly selecting 8 out of the 12 groups of concepts in MNIST-Add and providing the concepts corresponding to those 8 groups to all models at train-time (making sure the same group of 8 groups is selected across all baselines).

**CUB**  CUB is a visual bird-recognition task where each sample $\mathbf{x} \in [0, 1]^{3 \times 299 \times 299}$ represents an RGB image of a bird obtained from the CUB dataset [31]. Each of these images comes with a set of attribute annotations – e.g., "beak_type" and "bird_size" – as well as a downstream label $y$ representing the classification of one of 200 different bird types. In this work, we construct concept annotations $\mathbf{c}$ for this dataset using the same $k = 112$ bird attributes selected by Koh et al. [4] in their CBM work. These 112 concepts can be grouped into 28 groups of mutually exclusive concepts (e.g., "white wing", "black wing", etc). Following [4], we preprocess all of the images in this dataset by normalising, randomly flipping, and randomly cropping each image during training. We treat this dataset as a representative real-world task where one has access to a complete set of concept annotations at train-time.

**CUB-Incomp** CUB-Incomp is a concept-incomplete variant of the CUB task constructed by randomly selecting 25% of all concept groups in CUB and providing train-time annotations only for those concepts. For fairness, we ensure that the same groups are selected when training all baselines, resulting in a dataset with 7 groups of mutually exclusive concepts and 22 concepts.

**CelebA** The CelebA task is generated from samples in the Celebrity Attributes Dataset [32] using the task and concept annotations defined in [5]. For each image $\mathbf{x}$ in CelebA, we first select the 8 most balanced attributes $[a_1, \cdots a_8]$ from the 40 binary attributes available; we then build a vector of binary concept annotations $\mathbf{c}$ using only the 6 most balanced attributes $[a_1, \cdots, a_6]$, meaning that attributes $a_7$ and $a_8$ are not provided as concept annotations. To simulate a concept-incomplete task, we construct a task label $y$ via the base-10 representation of the number formed by $[a_1, \cdots, a_8]$. This results in a total of $l = 2^8 = 256$ classes. As in [5], we downsample every image to $(3, 64, 64)$ and randomly subsample images in CelebA's training set by selecting every $12^{\text{th}}$ sample.

## A.4 Model Architecture and Baselines Details

In this section, we describe the architectures and training hyperparameters used in our experiments.

### A.4.1 Model Architectures and Hyperparameters

To ensure a fair comparison across methods, we strive to use the exact same architectures for the concept encoder $g$ and label predictor $f$ across all baselines. Moreover, when possible, we follow the architectural choices selected in [5] to ensure that we are comparing our own IntCEM architecture against highly fine-tuned CEMs. The only tasks where this is not possible, as they were not a part of the original CEM evaluation, are the MNIST-based tasks. Below, we describe our architectural choices for $f$ and $g$ for each of our tasks. Moreover, for all tasks and models we include the choices of loss weights $\lambda_{\text{roll}}$ and $\lambda_{\text{concept}}$ where, for notational simplicity, we use $\lambda_{\text{concept}}$ to indicate the strength of the concept loss in both CEM-based models as well as in jointly trained CBMs. Finally, as in [5], for both CEMs and IntCEMs we use an embedding size of $m = 16$, a RandInt probability $p_{\text{int}} = 0.25$, and a single linear layer for all concept embedding generators $\{(\phi_i^-, \phi_i^+)\}_{i=1}^k$ and the shared scoring model $s$.

**MNIST-Based Tasks** For all MNIST-based tasks, the concept encoder $g$ begins by processing its input using a convolutional neural network backbone with 5 convolutional layers, each with sixteen $3 \times 3$ filters and a batch normalisation layer [43] before their nonlinear activation (a leaky-ReLU [44]), followed by a linear layer with $g_{\text{out}}$ outputs. When working with CEM-based models, the output of this backbone will be a $g_{\text{out}} = 128$, from which the concept embedding generators $\{(\phi_i^-, \phi_i^+)\}_{i=1}^k$ can produce embeddings $\{(\hat{\mathbf{c}}_i^-, \hat{\mathbf{c}}_i^+)\}_{i=1}^k$ and from which we can construct $g$. Otherwise, when working with CBM-based models, the output of this backbone will be the number of training concepts $g_{\text{out}} = k$, meaning that $g$ will be given by the backbone's output. For all models, we use as a label predictor $f$ a Multi-Layer Perceptron (MLP) with hidden layers $\{128, 128\}$ and leaky ReLU activations in between them. We use an MLP for $f$ rather than a simple linear layer, as it is commonly done, because we found that CBM-based models were unable to achieve high accuracy on these tasks unless a nonlinear model was used for $f$ (the same was not observed in CEM-based model as they have higher bottleneck capacity).

For CEMs, IntCEMs, and Joint CBM-Logit models, we use $\lambda_{\text{concept}} = 10$ across both MNIST-based tasks. This is only different for Joint CBM-Sigmoid were we noticed instabilities when setting this value to 10 (leading to large variances) and reduced it to $\lambda_{\text{concept}} = 5$ (reducing it lower than this value seem to decrease validation performance). All models are trained using a batch size of $B = 2048$ for a maximum of $E = 500$ epochs. To avoid models learning a majority class classifier, we weight each concept's cross entropy loss by its frequency and we use a weighted binary cross entropy loss for the task loss. Finally, for both MNIST tasks, the best performing IntCEM, determined by the model with the highest area under the intervention vs validation accuracy curve when $\lambda_{\text{roll}}$ varies through $\{5, 1, 0.1\}$, was obtained when $\lambda_{\text{roll}} = 1$.

**CUB-Based Tasks** As with our MNIST-based tasks, for all CUB-based tasks we construct a concept encoder $g$ by passing our input through a backbone model formed by a ResNet-34 [45] whose output linear layer has been modified to have $g_{\text{out}}$ activations. The output of this backbone is set to $g_{\text{out}} = k$

for all models and it is used to construct $g$ in the same way as described for the MNIST-based tasks. For all models, we use as a label predictor $f$ a linear layer.

For CEMs, IntCEMs, and Joint CBM-Logit models, we use $\lambda_{concept} = 5$ across both CUB-based tasks. The only exception is in `CUB-Incomp`, where we noticed a high variance in Joint CBM-Sigmoid when setting this value to 5 and reduced it to $\lambda_{concept} = 1$ as a response. All models are trained using a batch size of $B = 256$ for a maximum of $E = 300$ epochs and using a weight decay of $4 \times 10^{-5}$ as in [4]. As in the MNIST-based models, we scale each concept's binary cross entropy loss by a factor inversely proportional to its frequency to help the model learn the more imbalanced concepts. Finally, for both CUB tasks, the best performing IntCEM was obtained when $\lambda_{roll} = 1$.

**CelebA**   We use the same architecture and training configuration as in the CUB-based tasks for all `CelebA` models. The only differences between models trained in the CUB-tasks and those trained in `CelebA`, is that we set the batch size to $B = 512$ and train for a maximum of $E = 200$ epochs while setting the concept loss weight $\lambda_{concept} = 1$ for all joint CBMs and CEM-based models. When iterating over different values of $\lambda_{roll}$ for IntCEM, we found that $\lambda_{roll} = 5$ works best for this task.

### A.4.2   Training Hyperparameters

In our experiments, we endeavour to provide all models with a fair computational budget at train-time. To that end, we set a large maximum number of training epochs and stop training early based on each model's own validation loss. Here, we stop early if a model's validation loss does not decrease for 15 epochs. We monitor the validation loss by sampling 20% of the training set to make our validation set for each task. This validation set is also used to select the best-performing models whose results we report in Section 4. We always train with stochastic gradient descent with batch size $B$ (selected to well-utilise our hardware), initial learning rate $r_{initial}$, and momentum 0.9. Following the hyperparameter selection in reported in [5], we use $r_{initial} = 0.01$ for all tasks with the exception of the `CelebA` task where this is set to $r_{initial} = 0.005$. Finally, to help find better local minima in our loss function during training, we decrease the learning rate by a factor of 0.1 whenever the training loss plateaus for 10 epochs.

### A.4.3   IntCEM-Specific Parameters

As discussed in Section 3, IntCEM requires one to select a prior for its initial trajectory length and intervention mask, as well as an architecture to be used for the intervention policy model $\psi$. Below we describe who we set these values for all of our experiments. Later, we discuss the effect of some of these hyperparameters on our results in our ablation studies in Appendix A.6.

**Prior Selection**   In our experiments, to allow our prior to imitate how we intervene on IntCEMs at test-tine, we construct $p(\mu)$ by partitioning $\mathbf{c}$ into sets of mutually exclusive concepts and selecting concepts belonging to the same group with probability $p_{int} = 0.25$. Complementing this prior, we construct the intervention trajectory horizon's prior $p(T)$ using a discrete uniform prior $\text{Unif}(1, T_{max})$ where $T_{max}$ is annealed within $[2, 6]$ growing by a factor of 1.005 every training step. This was done to help the model stabilise its training by first learning meaningful concept representations before employing significant resources in improving its learnable policy $\psi$.

**Intervention Policy Architecture**   For all tasks, we parameterise IntCEM's learnable policy $\psi$ using a leaky-ReLU MLP with hidden layers $\{128, 128, 64, 64\}$ and Batch Normalisation layers in between them. When groups of mutually exclusive concepts are known at train-time (as in CUB-based tasks and MNIST-based tasks), we let $\psi$'s output distribution *operate over groups of mutually exclusive concepts* rather than individual concepts. This allows for a more efficient training process that better aligns with how these concepts will be intervened on at test-time. Finally, when training IntCEMs we use global gradient clipping to bound the global norm of gradients to 100. This helps us avoid exploding gradients [46] as we backpropagate gradients through our intervention trajectory.

### A.5   Computational Resource Usage

As mentioned in Sections 4 and 5, IntCEMs bear an additional overhead in their training times compared to traditional CEMs due to their need to sample intervention trajectories during training. In this section, we first study how training times vary across all of our baselines. Then we show that

IntCEM's training time overhead is amortised in practice if one considers the efficiency of IntCEM's learnt policy $\psi$ over existing state-of-the-art intervention policies.

Table A.2: Average wall-clock training times per epoch, epochs until convergence, and the number of parameters for baselines in the CUB task. All of these results were obtained by training 5 differently-initialised models for each baseline using the same hardware as specified in Appendix A.1.

| Model | Epoch Training Time (s) | Epochs To Convergence | Number of Parameters |
|---|---|---|---|
| Joint CBM-Sigmoid | $37.75 \pm 1.99$ | $225.67 \pm 27.18$ | $2.14 \times 10^7$ |
| Joint CBM-Logit | $36.76 \pm 0.70$ | $210.67 \pm 22.48$ | $2.14 \times 10^7$ |
| Sequential CBM | $31.08 \pm 1.17$ | $284.67 \pm 28.67$ | $2.14 \times 10^7$ |
| Independent CBM | $31.07 \pm 1.16$ | $284.67 \pm 28.67$ | $2.14 \times 10^7$ |
| CEM | $40.77 \pm 0.48$ | $174.00 \pm 34.88$ | $2.57 \times 10^7$ |
| IntCEM | $64.89 \pm 0.44$ | $224.00 \pm 26.77$ | $2.60 \times 10^7$ |

In Table A.2 we show the wall-clock times of a single training epoch for each of our baselines as well as the number of epochs it took the model to converge (as determined by our early stopping procedure defined in Appendix A.4). Our results indicate that, compared to traditional CEMs, IntCEMs take approximately $60\%$ longer than CEMs per training epoch while also taking approximately $\sim 20\%$ longer to converge (in terms of epochs). Nevertheless, we underscore three main observations from these results. First, the results in Table A.2 are very highly implementation-and-hardware-dependent, meaning that they should be taken with a grain of salt as they could vary widely across different implementations, batch sizes, and machines. Second, although on average we observe that IntCEMs take more epochs to converge than traditional CEMs, the variances we observe across different random initialisations are large. This means that these differences are not likely to be statistically significant; in fact, we observe some runs where IntCEMs converged faster than CEMs. Finally, as seen in Table A.3, these extra training costs may be amortised in practice when one considers that IntCEM's learnt policy is much faster than an equivalent competitive intervention policy like CooP. This implies that if these models are expected to be intervened frequently in practice, the overhead spanning from IntCEM's training procedure may result in better amortised computational resource usage over CEMs intervened on with the CooP policy.

Table A.3: Average wall-clock times taken to query the dynamic policies for trained IntCEMs across different tasks. Notice that all values are shown in microseconds (ms) and that these results can be highly hardware dependent, as they can be heavily affected if one runs a policy in a machine with GPU access. Nevertheless, all of these results were obtained by running all intervention policies using the same hardware as specified in Appendix A.1 and averaging over the entire test set.

| Dataset | Random (ms) | Learnt Policy $\psi$ (ms) | CooP (ms) | UCP (ms) | Skyline (ms) |
|---|---|---|---|---|---|
| MNIST-Add | $0.38 \pm 0.02$ | $0.37 \pm 0.02$ | $0.56 \pm 0.03$ | $0.40 \pm 0.04$ | $0.44 \pm 0.03$ |
| MNIST-Add-Incomp | $0.33 \pm 0.02$ | $0.32 \pm 0.01$ | $0.43 \pm 0.04$ | $0.31 \pm 0.00$ | $0.35 \pm 0.01$ |
| CUB | $2.91 \pm 0.13$ | $2.90 \pm 0.20$ | $4.77 \pm 0.10$ | $2.92 \pm 0.09$ | $4.54 \pm 0.09$ |
| CUB-Incomp | $3.19 \pm 0.39$ | $3.10 \pm 0.35$ | $5.38 \pm 0.85$ | $3.08 \pm 0.51$ | $4.73 \pm 0.27$ |
| CelebA | $0.70 \pm 0.01$ | $0.76 \pm 0.04$ | $0.73 \pm 0.02$ | $0.68 \pm 0.06$ | $0.76 \pm 0.06$ |

## A.6 Exploring IntCEM's Hyperparameter Sensitivity

In this section, we explore how our results reported in Section 4 change as we vary several of IntCEM's hyperparameters, specifically, $\lambda_{roll}$, $\lambda_{concept}$, $\gamma$, and $p_{int}$ and $T_{max}$.

We begin by exploring its sensitivity to its two loss weights, $\lambda_{roll}$ and $\lambda_{concept}$, and show that, although such parameters have an impact mostly on the unintervened accuracy of IntCEMs, our model is able to outperform CEMs once interventions are introduced throughout multiple different values for such weights. Then, we investigate how sensitive IntCEM is to its intervened prediction loss scaling factor $\gamma$ and show that its performance varies very little as $\gamma$ is modified. Finally, we explore how the hyperparameters we selected for IntCEM's prior ($p_{int}$ and $T_{max}$) affect its performance and describe similar results to those observed when varying IntCEM's loss weight hyperparameters.

For all the results we describe below, we train IntCEMs on the CUB task while fixing $\gamma = 1.1$, $\lambda_{concept} = 5$, $\lambda_{roll} = 1$, $p_{int} = 0.25$, and $T_{max} = 6$ unless any of these hyperparameters is the target

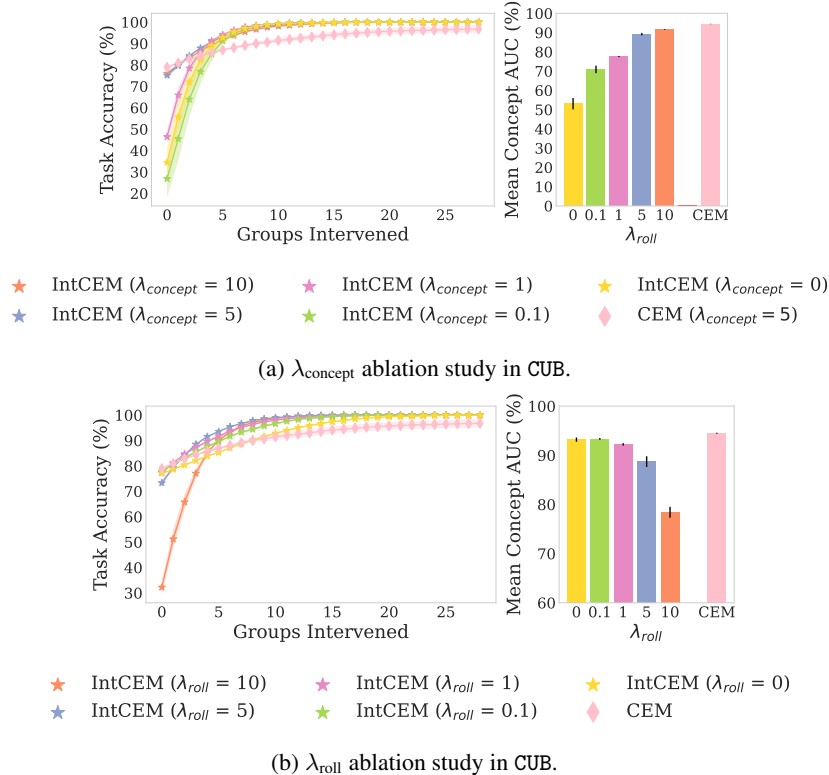

(a) $\lambda_{\text{concept}}$ ablation study in CUB.

(b) $\lambda_{\text{roll}}$ ablation study in CUB.

Figure A.1: Ablation studies in CUB for IntCEM's $\lambda_{\text{roll}}$ and $\lambda_{\text{concept}}$ loss weights. Notice that although initial unintervened performance may vary across different hyperparameters, in all instances we observe that IntCEMs outperform traditional CEMs (pink diamonds) after being intervened on.

of our ablation. Furthermore, we include as a reference the accuracy of a CEM trained in this same task. Finally, when showing intervention performance, all interventions in IntCEMs are made following their learnt policy $\psi$ while interventions in CEMs are made following the CooP policy (the best-performing policy for CEMs based on our experiments in Section 4).

### A.6.1 Rollout and Concept Loss Weights Ablations

In Figure A.1 we show the results of varying $\lambda_{\text{roll}}$ and $\lambda_{\text{concept}}$ in $\{0, 0.1, 1, 5, 10\}$ in the CUB task. Our ablation studies show that IntCEM's initial unintervened performance can vary as these two parameters are changed, particularly when $\lambda_{\text{concept}}$ is small or $\lambda_{\text{roll}}$ is large. Similarly, we observe that the average concept AUC of IntCEM decreases as $\lambda_{\text{concept}}$ is very small, as one would expect by $\lambda_{\text{concept}}$'s definition, and is affected as we increase the value of $\lambda_{\text{roll}}$, as described in Section 4.1. Nevertheless, and perhaps more importantly, we notice that for all hyperparameter configurations, IntCEM outperforms a traditional CEM's intervened task accuracy within five interventions. This is observed even when $\lambda_{\text{roll}} = 0$ or $\lambda_{\text{concept}} = 0$ and suggests that *our model's preconditioning to intervention receptiveness is robust to changes in its hyperparameters*. Moreover, we highlight that IntCEM's sensitivity to $\lambda_{\text{concept}}$ is not inherent to our architecture but rather inherent to any jointly trained CBMs and CEMs, as shown by Koh et al. [4] in their original CBM evaluation.

Looking at our $\lambda_{\text{roll}}$ ablation study, we observe that IntCEM's concept accuracy and unintervened task accuracy seem to decrease as $\lambda_{\text{roll}}$ increases. We hypothesise that happens as our model has a greater incentive to depend on interventions, and therefore can afford to make more concept and task mispredictions in the absence of interventions. Furthermore, we observe that the variance in IntCEM's unintervened task performance and concept performance as $\lambda_{\text{roll}}$ changes are less significant than that observed when $\lambda_{\text{concept}}$ changes, in fact achieving competitive accuracies for all values except when $\lambda_{\text{roll}} = 10$. This suggests that when fine-tuning IntCEM's loss hyperparameters using a validation set, as done in our experiments, it is recommended to leave $\lambda_{\text{roll}}$ within a small value (we found

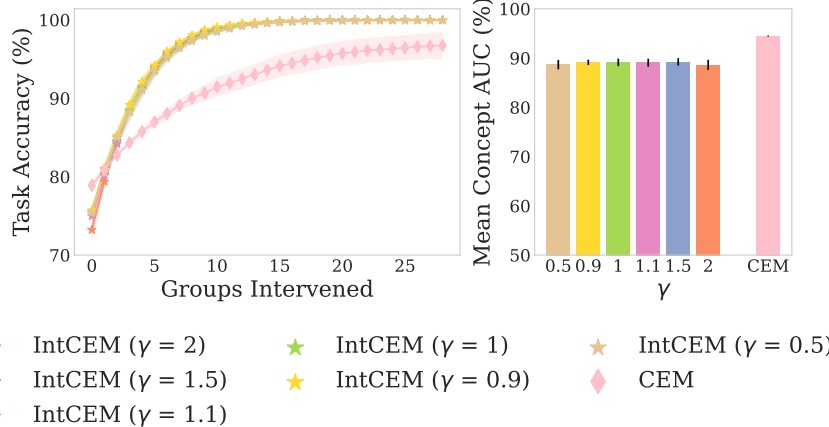

Figure A.2: Ablation study for the loss scaling factor $\gamma$. Our results show that IntCEM's task and concept accuracies are robust to changes of $\gamma$ within a reasonable range $[0.5, 2]$.

success with $\lambda_{\text{roll}} \in [0.1, 5]$) while spending more efforts fine-tuning $\lambda_{\text{concept}}$, a hyperparameter that unfortunately needs fine tuning for all jointly trained CBM-based architectures.

### A.6.2 Task Loss Scaling Factor

In Section 3, we defined $\gamma \in [1, \infty)$ as a hyperparameter that determines the penalty for mispredicting the task label $y$ with interventions $\mu^{(0)}$ versus mispredicting $y$ with interventions $\mu^{(T)}$ at the end of the trajectory. In all experiments reported in Section 4, we fixed $\gamma = 1.1$. In our ablation study for $\gamma$ (see Figure A.2), we show that IntCEM's receptiveness to interventions, and it mean concept AUC, is robust and not significantly affected as $\gamma$ varies within a reasonable range $[0.5, 2]$. In particular, and as expected, we observe that for a higher value of $\gamma$, the model has slightly lower unintervened accuracy while attaining slightly higher intervened accuracy after a few interventions. Nevertheless, these differences are small and suggest that our method is robust to minor variations in this hyperparameter.

### A.6.3 Prior Hyperparameters Ablation

IntCEM uses two prior distributions to sample trajectories at train-time: (i) $p(\mu_i) = \text{Bernoulli}(p_{\text{int}})$ on the initially sampled masks $\mu^{(0)} \in \{0, 1\}^k$ and (ii) $p(T) = \text{Unif}(1, T_{\text{max}})$ on the horizon, or length, of the intervention trajectory to be sampled. Our ablation studies on both $T_{\text{max}}$ and $p_{\text{int}}$ show that IntCEMs outperform CEMs when the number of interventions increases, regardless of the parameters used for $p_{\text{int}}$ and $T_{\text{max}}$. Nevertheless, we observe that when $T_{\text{max}}$ is small, e.g. $T_{\text{max}} = 1$, or $p_{\text{int}}$ is high, e.g. $p_{\text{int}} = 0.75$, IntCEM has a slower increase in performance as it is intervened. We believe this is a consequence of our model not having long-enough trajectories at train-time, therefore failing to condition the model to be receptive to long chains of interventions. This is because as $p_{\text{int}}$ is larger, the number of concepts initially intervened will be higher on expectation, leading to potentially less impactful concepts to select from at train-time. Similarly, as $T_{\text{max}}$ is lower, IntCEM's trajectories will be shorter and therefore it will not be able to explore the intervention space enough to learn better long-term trajectories. Therefore, based on these results our recommendation for these values, leveraging performance and train efficiency, is to set $T_{\text{max}}$ to a small value in $[5, 10]$ (we use $T_{\text{max}} = 6$ in our experiments) and $p_{\text{int}}$ to a value within $[0, 0.5]$ (we use $p_{\text{int}} = 0.25$ in our experiments).

### A.7 Additional Results and Details on Intervention Policy Evaluation

In this section, we provide further details on the set-up of our intervention policy experiments and include additional results on the impact of intervening on IntCEMs and our baselines.

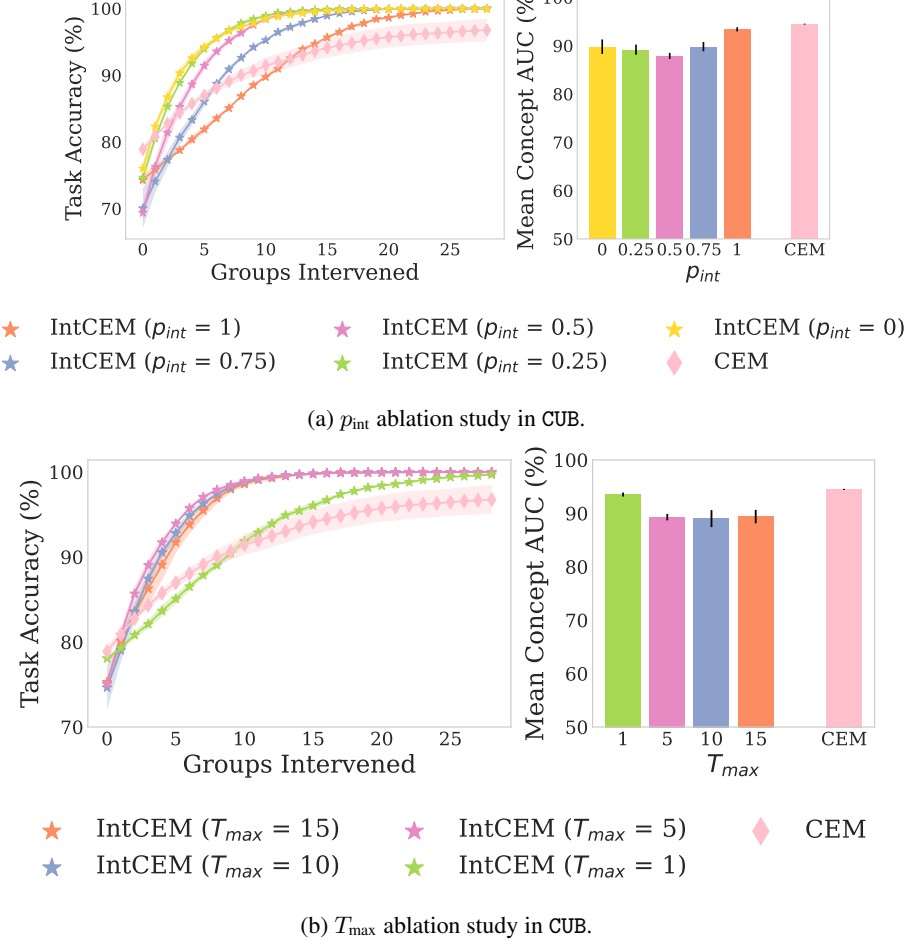

(a) $p_{\text{int}}$ ablation study in `CUB`.

(b) $T_{\max}$ ablation study in `CUB`.

Figure A.3: Ablation studies in `CUB` for IntCEM's priors' hyperparameters. Notice that, as in previous ablations, although initial unintervened performance may vary across different hyperparameters, regardless of hyperparameter selection IntCEM outperforms traditional CEMs (shown in pink diamonds above) within ∼12 concept groups being intervened.

### A.7.1 Intervention Policy Details

**Intervening on Logit CBMs** Because Joint CBM-Logits use log-probabilities in their bottleneck rather than sigmoidal zero-one probabilities, intervening on their bottlenecks is slightly different than intervening on Joint CBM-Sigmoidal models. As in [4], we avoid setting intermediate concepts to out-of-distribution values by setting $\hat{c}_i$ to its empirical 95-th percentile value (computed over the training distribution) when we intervene on concept $c_i$ and want to indicate that its ground truth value is $\tilde{c}_i = 1$. Similarly, when we intervene on concept $c_i$ and want to indicate that its ground truth value is $\tilde{c}_i = 0$, we set it to its empirical 5-th percentile value (computed over its training distribution). This allows us to extend the intervention formulation in Equation (1) to CBMs with logit bottlenecks by letting $\hat{\mathbf{c}}_i^+$ be the 95-th percentile value of the empirical training distribution of $\hat{c}_i$ and letting $\hat{\mathbf{c}}_i^-$ be the 5-th percentile value of the same distribution.

**CooP** CooP selects the best next concept to intervene on by considering a weighted sum of (i) an uncertainty measurement of each concept's predicted probability (weighted by hyperparameter $\alpha$), (ii) the expected change in the probability of the currently predicted class (weighted by hyperparameter $\beta$), and (iii) the cost of acquiring each concept (weighted by hyperparameter $\gamma_{\text{coop}}$). For simplicity, in this work, we treat all concepts as having the same acquisition cost; i.e., we set $\gamma_{\text{coop}} = 0$ and only fine-tune $\alpha$ and $\beta$. We leave the exploration of variants of our method which consider different acquisition costs for future work.

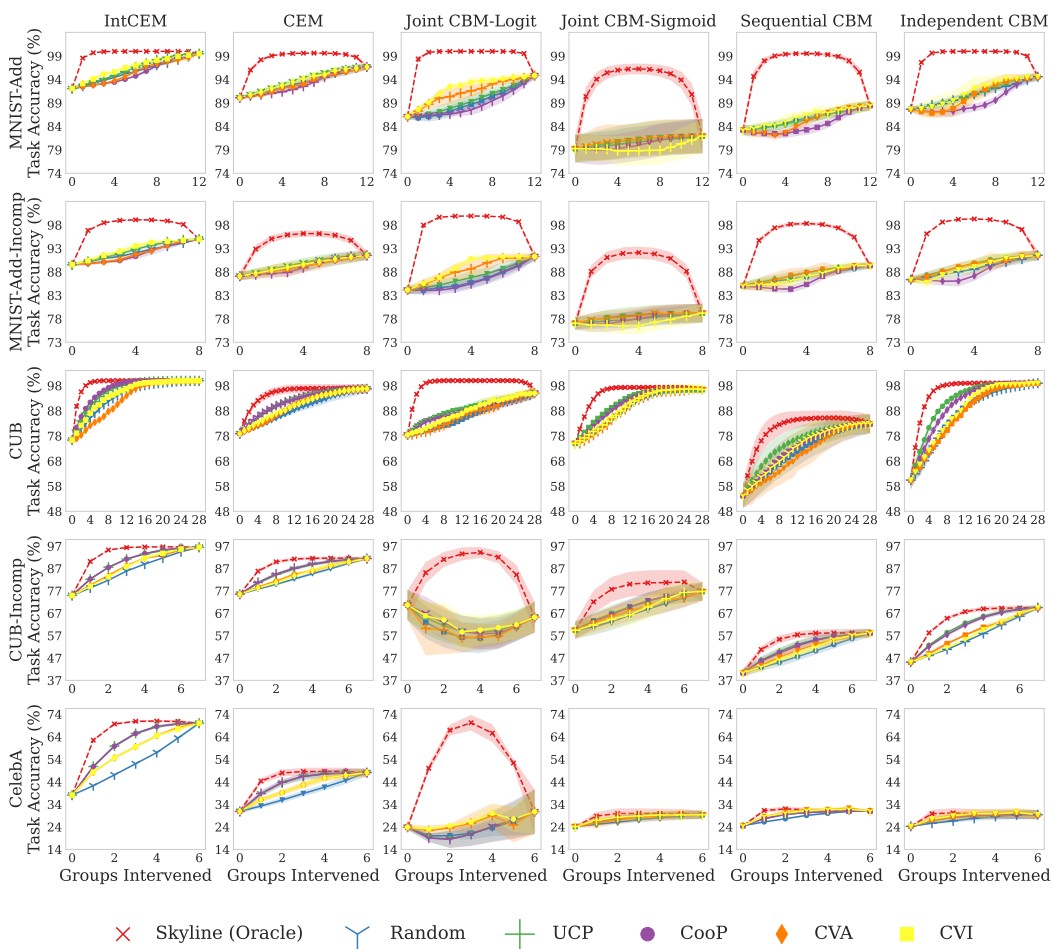

Figure A.4: Task accuracy when intervening with different policies (colours) on different methods (columns) and datasets (rows). We observe that, across all tasks and datasets, IntCEMs outperform all other baselines when intervened on. This difference is particularly sharp on the more complicated datasets such as `CUB-Incomp` and `CelebA`.

When using CooP for any of our baselines, we select $\alpha$ and $\beta$ by performing a grid search over $\alpha \in \{0.1, 1, 10, 100\}$ and $\beta \in \{0.1, 1, 10, 100\}$ and selecting the pair of hyperparameters that give us the best area under the validation intervention curve. Given that this process can be costly (in our runs it may take up to $\sim$45 minutes to run), we estimate the area under the validation intervention curve by checking CooP performance after intervening with $1\%, 5\%, 25\%$, and $50\%$ of the available concept groups for a given task.

**Behavioural Cloning Policy** To learn a behavioural cloning policy which imitates *Skyline* through demonstrations of the latter policy, we train a leaky-ReLU MLP with hidden layers $\{256, 128\}$ that maps a previous bottleneck $\hat{\mathbf{c}}$ and a mask of previously intervened concepts $\mu$ to $k$ outputs indicating the log-probabilities of selecting each concept as the next intervention. This model is trained by generating $5,000$ demonstrations $((\hat{\mathbf{c}}, \mu), \eta_{\text{sky}})$ of "Skyline".

Each demonstration is formed by (1) selecting a training sample $\mathbf{x}$ from the task's training set uniformly at random, (2) generating an initial intervention mask $\mu$ by selecting, uniformly at random, $l \sim \text{Unif}(0, k)$ concepts to intervene on, (3) producing an initial bottleneck $\hat{\mathbf{c}} = \tilde{g}(\mathbf{x}, \mu, \mathbf{c})$ using concept encoder $g$ of the model we will intervene on and $\mathbf{x}$'s corresponding concept labels $\mathbf{c}$, and (4) generating a target concept intervention $\eta_{\text{sky}}$ by calling the *Skyline* policy with inputs $(\hat{\mathbf{c}}, \mu)$. This BC model is then trained to minimise the cross entropy loss between its predicted probability distribution

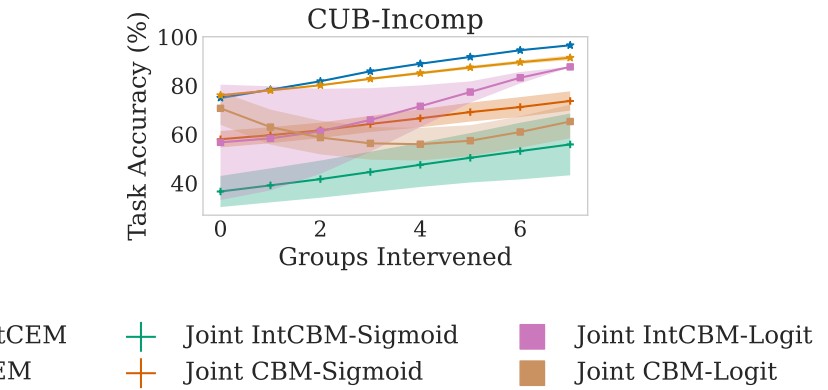

Figure A.5: Test accuracy for traditional CBMs trained with IntCEM's loss functions (*IntCBMs*) on the `CUB-Incomp` dataset. Our results suggest that although there may be merit to incorporating IntCEM's loss into traditional CBMs, their use of scalar concept representation severely limits their stability and unintervened performance.

and $\eta_{\text{sky}}$ for 100 epochs using stochastic gradient descent with learning rate 0.01, batch size 256, weight decay $4 \times 10^{-5}$, and momentum 0.9.

We highlight that when selecting concepts from both the BC policy and the learnt policy $\psi$ at test-time, **we deterministically select the concept with the maximum log-probability that has not yet been selected by** $\mu$.

### A.7.2 Additional Results

As discussed in Section 4, in Figure A.4 we observe that IntCEMs achieve significantly higher task accuracies across all tasks under interventions, regardless of the intervention policy used. We notice, however, that the gap between the optimal intervention policy and IntCEM's best-performing policy tends to be larger in the simpler datasets. Further work can investigate the origins of this gap as well as mechanisms to bridge it.

### A.8 Extending IntCEM's Loss to Traditional CBMs

Because IntCEM's loss function only assumes that one can intervene on a given model at train-time, this loss function is general enough to be applicable to traditional Joint CBMs. In Figure A.5 we show the results of intervening on joint CBMs whose training objectives have been modified to include IntCEM's $\mathcal{L}_{\text{roll}}$ and $\mathcal{L}_{\text{pred}}$ terms (*IntCBMs*). These results show that IntCBMs, in particular when using a logit bottleneck, can achieve a higher test accuracy than their traditional CBM counterparts when presented with a large number of test-time interventions. Nevertheless, we note the following limitations: (i) we observe that for IntCBMs with sigmoidal bottlenecks, there is no improvement regardless of the use of interventions, and in fact, adding $\mathcal{L}_{\text{pred}}$ and $\mathcal{L}_{\text{roll}}$ to traditional CBMs can in fact lead to worse performance; (ii) even in IntCBMs with logit bottlenecks whose intervened performance outperforms their CBM counterparts, we observe a high variance, especially when the number of intervened concepts is small; and (iii) in comparison to IntCEMs, the improvements of our training loss in IntCBMs with logit activations are still significantly underperforming when receiving little-to-none interventions. We believe that these observations arise for two core reasons: first, using concept embeddings allows for a better flow of information between the concept encoder and the label predictor, leading to embedding-based models being more robust to concept incompleteness than their scalar-based counterparts (especially with respect to sigmoidal bottleneck). Second, using embeddings rather than scalar concept representations enable gradients to flow from the label predictor into the concept encoder *even after interventions are made*, something that is not possible with traditional CBMs as setting a bottleneck activation to a fixed value (e.g., 0 or the 5-th percentile of the activation) is a gradient-blocking operation. This also leads to significantly more stable gradient propagation throughout an intervention trajectory, explaining the high variance in IntCBMs. These results suggest that using high-dimensional embedding representations for concepts may be a crucial

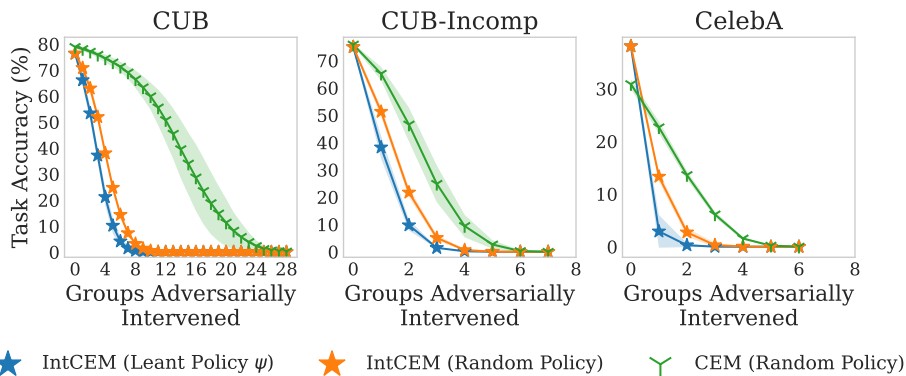

Figure A.6: Task accuracy for IntCEMs and CEMs as we adversarially intervene on an increasingly large number of concept groups (by intentionally selecting the wrong concept label). For IntCEMs we show interventions following its learnt intervention policy $\psi$ as well as a Random policy. For CEMs, all interventions are done following a Random intervention policy.

component for these models to be receptive to interventions, and further, simply including a higher train-time penalty when mispredicting a label after an intervention trajectory is not enough for a model to become competitive with and without interventions.

## A.9 Effect of Adversarial Interventions

In their evaluation, Espinosa Zarlenga et al. [5] observed that CEMs were somewhat more robust to adversarial interventions (i.e., interventions that are intentionally incorrect) compared to existing CBM architectures – where robustness was defined as being able to withstand a certain number of wrong interventions before significantly dropping in performance. This suggests that CEMs may have a form of error correction in their embeddings that allow the label predictor to correct intervention mistakes. In contrast, as posited in our discussion in Section 5, by encouraging IntCEMs to be receptive to test-time interventions, we are also incentivising them to be receptive to *all forms* of interventions, even if those interventions are adversarial in nature. This is because our model's explicit train-time intervention loss, which encourages IntCEMs to positively respond to test-time interventions, has the underlying assumption that interventions, when provided at train-time or test-time, are correct. Hence, we hypothesise that the same error correction observed in CEMs will not be seen in IntCEMs, leading our models' performance to significantly drop when interventions are wrong. We find evidence supporting this hypothesis when performing adversarial interventions across all our baselines on the CUB, CUB-Incomp, and CelebA tasks. Specifically, as seen in Figure A.6, we observe that an IntCEM's performance drastically drops when receiving only a handful of adversarial interventions, with this result being even more damaging if we follow its learnt intervention policy $\psi$. These results suggest that a very interesting and impactful direction for future work can be to explore mechanisms to introduce error correction in IntCEM-like models.

## A.10 Leakage in IntCEM's Learnt Representations

Concept leakage [36, 10] is a known issue in CBMs where a CBM's concept encoder learns to encode unnecessary information in its learnt concept representations in order to bypass information from the input to its label predictor. It has been empirically shown that leakage may lead to less interpretable concept representations [36] and to detrimental interventions on CBMs [10]. The latter is a consequence of a CBM's label predictor learning to rely on information that accidentally leaked into the concept representations, information that is necessarily destroyed at intervention-time when an expert overwrites a bottleneck activation with its ground truth value. Nevertheless, the same is not true in embedding-based architectures, such as CEMs and IntCEMs, where interventions simply involve "swapping" an embedding with one that can still carry leaked information into the label predictor (see Equation 1). Hence, in this section we hypothesize that higher leakage may, in fact, be a contributor to IntCEMs' better performance when being intervened on as the model may learn to take better advantage of this leakage to learn to be more receptive to interventions at test-time.

Table A.4: Oracle Impurity Score (OIS) for all jointly-trained baselines across all tasks. Higher OIS values indicate higher leakage in a model's learnt concept representations.

| | Dataset | IntCEM | CEM | Joint CBM-Sigmoid | Joint CBM-Logit |
|---|---|---|---|---|---|
| OIS (%) | MNIST-Add | 30.97 ± 0.29 | 28.25 ± 0.44 | 13.14 ± 0.27 | 20.96 ± 0.02 |
| | MNIST-Add-Incomp | 36.73 ± 0.23 | 34.09 ± 0.47 | 13.28 ± 0.27 | 26.12 ± 0.08 |
| | CUB | 45.86 ± 0.29 | 42.54 ± 2.30 | 21.01 ± 0.58 | 41.66 ± 0.49 |
| | CUB-Incomp | 38.79 ± 1.41 | 42.13 ± 3.48 | 27.81 ± 2.02 | 30.32 ± 3.09 |
| | CelebA | 50.87 ± 4.07 | 40.63 ± 4.83 | 29.65 ± 16.51 | 24.11 ± 09.30 |

We evaluate our hypothesis by measuring the Oracle Impurity Score (OIS) [10] of concept representations learnt by CBMs, CEMs, and IntCEMs across all tasks. This score, between 0 and 1, measures how much extra information, on average, each learnt concept representation captures from other possibly unrelated concepts (with higher scores representing higher impurities and, therefore, more leakage between concepts). Our results, shown in Table A.4, show that not only is there significantly more leakage in CEMs compared to CBMs, as one would expect given their higher capacity, but IntCEM's embeddings are capturing more impurities than CEM's embeddings across all tasks but one, providing evidence towards our hypothesis. This preliminary study suggests that contrary to common assumptions, leakage may not always be undesired and could be a healthy byproduct of more expressive concept representations in models that accommodate such expressivity. Nevertheless, this phenomenon may also potentially lead to IntCEM's concept representations being less interpretable than those in CEMs when such leakage is unaccounted for. Therefore, we encourage future work to explore leakage's positive and negative consequences in embedding-based concept models to design better inductive biases for effective interventions.

## A.11 Frequency of Selected Concepts by IntCEM's policy

In Figure A.7 we show the frequencies of concept groups selected by IntCEM's $\psi$ policy when intervening on the validation set of CUB. To study how much this policy changes across different initialisations and random seeds for the same hyperparameters and models, we show these histograms across multiple instances of identically trained IntCEMs with different random seeds. We notice that although, in most cases, concepts such as "upperparts colour" and "size" seem to be highly preferred in the early steps of intervention, there seems to be a relatively large variance across multiple seeds. This may suggest that there are several equally informative or good concepts one may request at a given time to obtain similar improvements, something that we leave for future work to explore.

## A.12 Tabulation of Results

This section presents Tables A.5, A.6, and A.7, a set of tabular summaries of our results introduced in Section 4. These summaries show the precise means and standard deviations (computed over five different random seeds) of test-time accuracies after interventions for all baselines and tasks when deploying various intervention policies. They represent tabular versions of the results presented in Figures 3, 4, and 5, respectively.

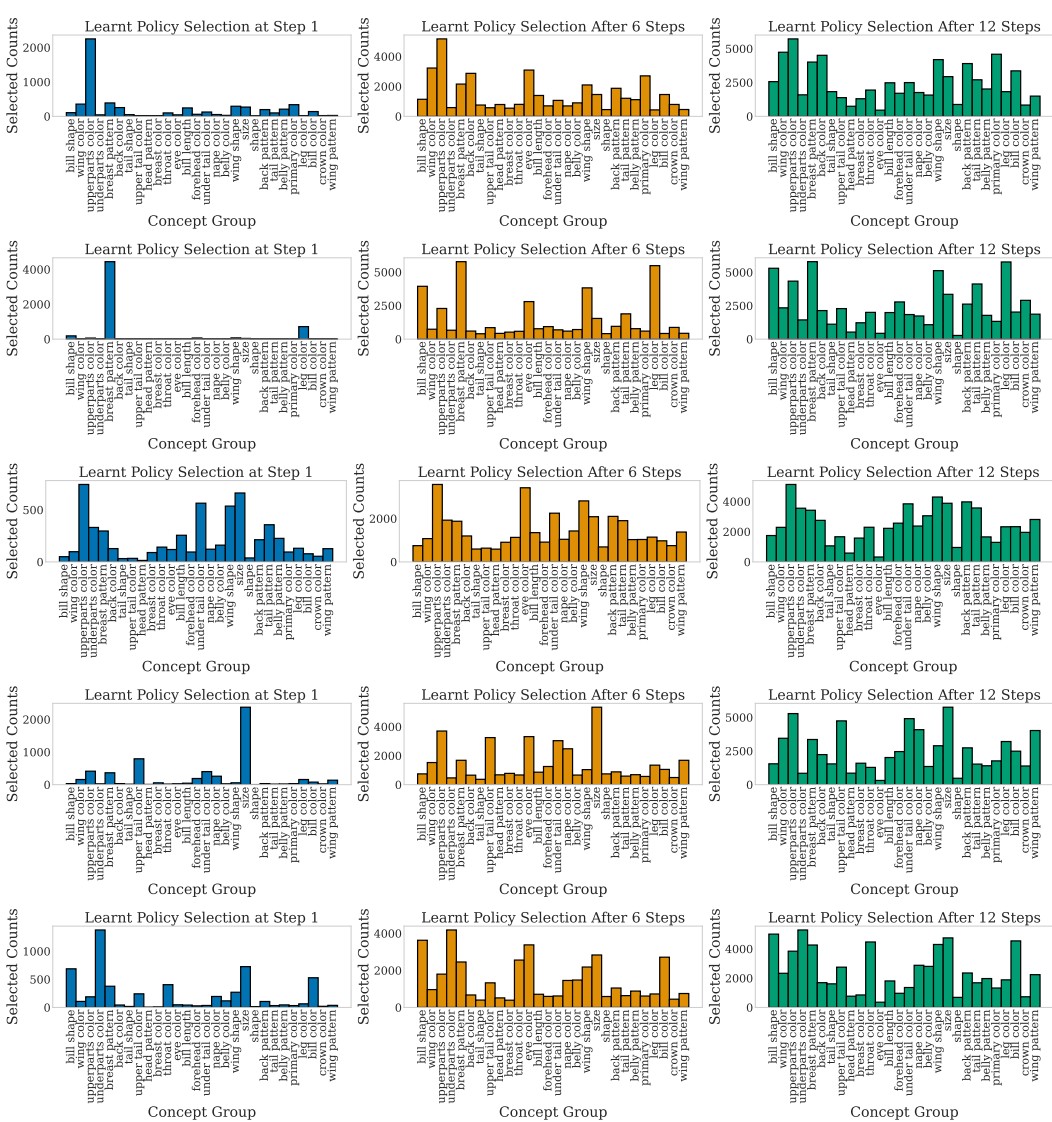

Figure A.7: Frequency of selected concept groups by IntCEM's learnt policy $\psi$ after a given number of interventions in CUB's validation set. Each column represents the distribution of intervened groups after a fixed number of interventions. Similarly, each row represents a different initialisation and seed for IntCEM before training.

Table A.5: Tabular summary of Figure 3. Task predictive performance (accuracy or AUC in %) across all tasks and baselines when intervening on a fixed fraction of the set of available concepts at test time (indicated on the left-hand side of the dataset names). Except for "IntCEM (Learnt Policy $\psi$)", all interventions use a random intervention policy. We write in bold the best performance results across baselines following a random intervention policy.

| | Dataset | IntCEM (Random) | CEM | Joint CBM-Sigmoid | Joint CBM-Logit | Ind. CBM | Seq. CBM | IntCEM (Learnt Policy $\psi$) |
|---|---|---|---|---|---|---|---|---|
| 25% | MNIST-Add | **93.78 ± 0.09** | 91.66 ± 0.41 | 86.47 ± 1.11 | 79.86 ± 3.12 | 84.11 ± 1.09 | 88.88 ± 0.67 | 94.75 ± 0.28 |
| | MNIST-Add-Incomp | **90.88 ± 0.25** | 88.22 ± 0.92 | 84.68 ± 0.83 | 77.59 ± 1.57 | 86.06 ± 0.77 | 87.53 ± 0.74 | 91.69 ± 0.39 |
| | CUB | **88.53 ± 0.23** | 84.08 ± 0.36 | 83.18 ± 0.22 | 81.76 ± 0.90 | 79.96 ± 1.57 | 65.40 ± 5.38 | 94.10 ± 0.49 |
| | CUB-Incomp | **78.34 ± 0.34** | 77.83 ± 0.33 | 60.94 ± 3.66 | 62.97 ± 7.19 | 48.16 ± 0.62 | 42.78 ± 1.63 | 81.71 ± 0.23 |
| | CelebA | **42.25 ± 0.36** | 33.27 ± 0.94 | 25.16 ± 0.58 | 20.02 ± 1.27 | 25.63 ± 0.55 | 26.21 ± 0.32 | 47.63 ± 1.49 |
| 50% | MNIST-Add | **95.70 ± 0.11** | 93.36 ± 0.51 | 88.66 ± 1.20 | 80.69 ± 3.30 | 85.71 ± 0.94 | 90.87 ± 0.83 | 96.71 ± 0.20 |
| | MNIST-Add-Incomp | **92.07 ± 0.21** | 89.13 ± 0.96 | 85.73 ± 1.12 | 77.96 ± 1.73 | 86.87 ± 0.74 | 88.58 ± 0.95 | 92.98 ± 0.31 |
| | CUB | **96.14 ± 0.35** | 89.66 ± 1.53 | 92.28 ± 0.74 | 86.37 ± 1.18 | 92.19 ± 0.77 | 74.24 ± 5.68 | 99.17 ± 0.14 |
| | CUB-Incomp | **85.86 ± 0.29** | 82.32 ± 0.58 | 66.12 ± 3.81 | 56.30 ± 6.71 | 54.35 ± 0.35 | 47.62 ± 1.85 | 90.69 ± 0.30 |
| | CelebA | **52.01 ± 0.45** | 38.82 ± 1.32 | 27.19 ± 1.17 | 21.20 ± 2.74 | 27.79 ± 1.07 | 29.11 ± 0.40 | 62.01 ± 1.21 |
| 75% | MNIST-Add | **97.51 ± 0.08** | 95.01 ± 0.57 | 91.24 ± 1.05 | 81.31 ± 3.44 | 86.97 ± 0.96 | 92.58 ± 0.82 | 98.22 ± 0.09 |
| | MNIST-Add-Incomp | **93.48 ± 0.25** | 90.35 ± 0.96 | 88.09 ± 1.03 | 78.62 ± 1.90 | 88.15 ± 0.68 | 90.09 ± 0.91 | 94.18 ± 0.24 |
| | CUB | **98.98 ± 0.14** | 93.95 ± 1.95 | 95.73 ± 1.00 | 91.18 ± 1.61 | 96.97 ± 0.53 | 79.81 ± 4.96 | 99.82 ± 0.08 |
| | CUB-Incomp | **91.74 ± 0.19** | 86.93 ± 0.61 | 71.54 ± 3.93 | 57.39 ± 6.36 | 61.92 ± 0.52 | 53.00 ± 1.76 | 94.96 ± 0.26 |
| | CelebA | **56.99 ± 0.37** | 41.59 ± 1.39 | 28.04 ± 1.42 | 23.70 ± 3.82 | 28.63 ± 1.48 | 30.04 ± 0.48 | 66.48 ± 1.10 |
| 100% | MNIST-Add | **99.51 ± 0.04** | 96.68 ± 0.68 | 94.84 ± 0.75 | 81.92 ± 3.65 | 88.43 ± 1.23 | 94.58 ± 0.81 | 99.51 ± 0.04 |
| | MNIST-Add-Incomp | **94.99 ± 0.11** | 91.53 ± 1.01 | 91.28 ± 0.81 | 79.21 ± 1.98 | 89.36 ± 0.69 | 91.54 ± 1.03 | 94.99 ± 0.11 |
| | CUB | **99.90 ± 0.04** | 96.75 ± 1.72 | 96.42 ± 1.24 | 95.03 ± 1.95 | 98.97 ± 0.35 | 82.81 ± 3.72 | 99.90 ± 0.04 |
| | CUB-Incomp | **96.52 ± 0.19** | 91.47 ± 0.67 | 76.72 ± 4.14 | 65.25 ± 6.88 | 69.54 ± 1.05 | 58.19 ± 1.85 | 96.52 ± 0.19 |
| | CelebA | **70.02 ± 0.62** | 48.10 ± 1.72 | 29.38 ± 1.80 | 30.75 ± 9.89 | 29.43 ± 2.02 | 31.04 ± 0.22 | 70.02 ± 0.62 |

Table A.6: Tabular summary of Figure 4 including performance in other tasks as shown in Appendix A.7 but excluding performance when using the "Random" policy (see Table A.5 for those results). Here, we show the task accuracy of IntCEMs and CEMs when intervening with different test-time policies as we vary the fraction of concept groups we intervene on (indicated on the left-hand side of the dataset names).

| | Dataset | IntCEM (UCP) | CEM (UCP) | IntCEM (CooP) | CEM (CooP) | IntCEM (CVA) | CEM (CVA) | IntCEM (CVI) | CEM (CVI) | IntCEM (Skyline) | CEM (Skyline) |
|---|---|---|---|---|---|---|---|---|---|---|---|
| 25% | MNIST-Add | 94.07 ± 0.22 | 92.17 ± 0.46 | 93.16 ± 0.14 | 91.06 ± 0.41 | 93.14 ± 0.12 | 91.39 ± 0.65 | 95.14 ± 0.37 | 92.13 ± 0.70 | 99.93 ± 0.01 | 99.05 ± 0.26 |
| | MNIST-Add-Incomp | 91.03 ± 0.38 | 88.53 ± 0.92 | 90.09 ± 0.34 | 87.63 ± 0.84 | 90.11 ± 0.30 | 88.21 ± 1.28 | 91.52 ± 0.68 | 88.37 ± 1.12 | 98.48 ± 0.08 | 95.06 ± 0.77 |
| | CUB | 93.37 ± 0.20 | 88.56 ± 0.69 | 94.68 ± 0.25 | 89.09 ± 0.75 | 86.91 ± 1.05 | 84.81 ± 0.55 | 89.44 ± 1.03 | 86.01 ± 0.50 | 99.76 ± 0.05 | 95.25 ± 1.81 |
| | CUB-Incomp | 82.20 ± 0.11 | 80.40 ± 0.26 | 82.34 ± 0.24 | 80.77 ± 0.47 | 79.49 ± 0.57 | 78.68 ± 0.51 | 79.08 ± 0.37 | 78.35 ± 0.53 | 90.16 ± 0.31 | 85.83 ± 0.72 |
| | CelebA | 50.82 ± 0.41 | 38.67 ± 1.26 | 50.72 ± 0.56 | 38.83 ± 1.36 | 48.83 ± 0.34 | 36.04 ± 1.14 | 48.28 ± 0.34 | 36.04 ± 1.14 | 62.48 ± 0.54 | 44.34 ± 1.27 |
| 50% | MNIST-Add | 96.60 ± 0.13 | 94.22 ± 0.55 | 94.74 ± 0.27 | 92.54 ± 0.44 | 95.70 ± 0.42 | 93.05 ± 0.83 | 96.97 ± 0.83 | 94.17 ± 0.78 | 100.00 ± 0.00 | 99.61 ± 0.15 |
| | MNIST-Add-Incomp | 92.82 ± 0.45 | 89.97 ± 0.98 | 91.31 ± 0.59 | 88.89 ± 0.98 | 91.70 ± 0.46 | 89.19 ± 1.31 | 93.53 ± 0.64 | 89.59 ± 0.92 | 99.11 ± 0.18 | 96.17 ± 0.69 |
| | CUB | 98.52 ± 0.19 | 93.70 ± 1.43 | 99.27 ± 0.16 | 93.60 ± 1.39 | 96.35 ± 0.70 | 91.71 ± 1.47 | 97.45 ± 0.40 | 91.49 ± 1.38 | 99.92 ± 0.01 | 96.87 ± 1.71 |
| | CUB-Incomp | 91.21 ± 0.22 | 86.82 ± 0.73 | 91.25 ± 0.13 | 87.17 ± 0.74 | 88.31 ± 0.30 | 84.42 ± 0.59 | 87.90 ± 0.73 | 83.76 ± 1.07 | 96.38 ± 0.28 | 91.23 ± 0.73 |
| | CelebA | 65.47 ± 0.59 | 46.29 ± 1.49 | 65.47 ± 0.57 | 46.48 ± 1.59 | 59.60 ± 0.38 | 42.61 ± 1.38 | 59.54 ± 0.39 | 42.61 ± 1.38 | 70.84 ± 0.47 | 48.56 ± 1.63 |
| 75% | MNIST-Add | 98.78 ± 0.04 | 95.93 ± 0.65 | 97.99 ± 0.46 | 95.24 ± 0.88 | 97.80 ± 0.67 | 95.24 ± 0.88 | 98.43 ± 0.44 | 95.93 ± 0.69 | 100.00 ± 0.00 | 99.54 ± 0.19 |
| | MNIST-Add-Incomp | 94.42 ± 0.22 | 91.06 ± 1.02 | 93.62 ± 0.82 | 90.83 ± 1.03 | 93.64 ± 0.15 | 90.43 ± 1.01 | 94.53 ± 0.23 | 90.74 ± 0.94 | 98.87 ± 0.15 | 95.81 ± 0.74 |
| | CUB | 99.75 ± 0.07 | 95.93 ± 1.56 | 99.88 ± 0.04 | 95.22 ± 1.55 | 99.18 ± 0.23 | 95.17 ± 1.72 | 99.74 ± 0.04 | 95.22 ± 1.55 | 99.93 ± 0.02 | 96.96 ± 1.70 |
| | CUB-Incomp | 95.04 ± 0.31 | 90.03 ± 0.87 | 95.37 ± 0.31 | 90.24 ± 0.79 | 92.92 ± 0.38 | 88.35 ± 0.85 | 93.81 ± 0.95 | 88.37 ± 1.36 | 96.68 ± 0.25 | 91.63 ± 0.69 |
| | CelebA | 68.54 ± 0.67 | 47.43 ± 1.63 | 68.54 ± 0.66 | 47.51 ± 1.65 | 64.46 ± 0.47 | 45.65 ± 1.46 | 64.46 ± 0.47 | 45.65 ± 1.46 | 70.95 ± 0.52 | 48.61 ± 1.66 |
| 100% | MNIST-Add | 99.51 ± 0.04 | 96.68 ± 0.68 | 99.51 ± 0.04 | 96.68 ± 0.68 | 99.51 ± 0.04 | 96.68 ± 0.68 | 99.51 ± 0.04 | 96.68 ± 0.68 | 99.51 ± 0.04 | 96.68 ± 0.68 |
| | MNIST-Add-Incomp | 94.99 ± 0.11 | 91.53 ± 1.01 | 94.99 ± 0.11 | 91.53 ± 1.01 | 94.99 ± 0.11 | 91.53 ± 1.01 | 94.99 ± 0.11 | 91.53 ± 1.01 | 94.99 ± 0.11 | 91.53 ± 1.01 |
| | CUB | 99.90 ± 0.04 | 96.75 ± 1.72 | 99.90 ± 0.04 | 96.75 ± 1.72 | 99.90 ± 0.04 | 96.75 ± 1.72 | 99.90 ± 0.04 | 96.75 ± 1.72 | 99.90 ± 0.04 | 96.75 ± 1.72 |
| | CUB-Incomp | 96.52 ± 0.19 | 91.47 ± 0.67 | 96.52 ± 0.19 | 91.47 ± 0.67 | 96.52 ± 0.19 | 91.47 ± 0.67 | 96.52 ± 0.19 | 91.47 ± 0.67 | 96.52 ± 0.19 | 91.47 ± 0.67 |
| | CelebA | 70.02 ± 0.62 | 48.10 ± 1.72 | 70.02 ± 0.62 | 48.10 ± 1.72 | 70.02 ± 0.62 | 48.10 ± 1.72 | 70.02 ± 0.62 | 48.10 ± 1.72 | 70.02 ± 0.62 | 48.10 ± 1.72 |

Table A.7: Tabular summary of Figure 5. Here, we show task performance when intervening on IntCEMs following test-time policies $\psi$, *CooP*, *Random*, and *BC-Skyline*. The fraction of concept intervened on for each dataset is shown on the left-hand-side of the table.

| | Dataset | Random | CooP | Learnt Policy | BC-Skyline | IntCEM no $\psi$ (Random) |
|---|---|---|---|---|---|---|
| 25% | MNIST-Add | 93.78 ± 0.09 | 93.16 ± 0.14 | 94.75 ± 0.28 | 93.20 ± 0.19 | 93.81 ± 0.18 |
| | MNIST-Add-Incomp | 90.88 ± 0.25 | 90.09 ± 0.34 | 91.69 ± 0.39 | 90.18 ± 0.35 | 90.98 ± 0.18 |
| | CUB | 88.53 ± 0.23 | 94.68 ± 0.25 | 94.10 ± 0.49 | 88.43 ± 0.29 | 88.46 ± 0.46 |
| | CUB-Incomp | 78.34 ± 0.34 | 82.34 ± 0.24 | 81.71 ± 0.23 | 79.06 ± 0.41 | 78.15 ± 0.30 |
| | CelebA | 42.25 ± 0.36 | 50.72 ± 0.56 | 47.63 ± 1.49 | 48.47 ± 0.47 | 33.45 ± 0.58 |
| 50% | MNIST-Add | 95.70 ± 0.11 | 94.74 ± 0.27 | 96.71 ± 0.20 | 94.56 ± 0.47 | 95.69 ± 0.11 |
| | MNIST-Add-Incomp | 92.07 ± 0.21 | 91.31 ± 0.59 | 92.98 ± 0.31 | 91.56 ± 0.43 | 92.13 ± 0.16 |
| | CUB | 96.14 ± 0.35 | 99.27 ± 0.16 | 99.17 ± 0.14 | 96.28 ± 0.19 | 95.89 ± 0.40 |
| | CUB-Incomp | 85.86 ± 0.29 | 91.25 ± 0.13 | 90.69 ± 0.30 | 86.74 ± 0.41 | 85.31 ± 0.41 |
| | CelebA | 52.01 ± 0.45 | 65.47 ± 0.57 | 62.01 ± 1.21 | 59.95 ± 0.84 | 40.71 ± 0.56 |
| 75% | MNIST-Add | 97.51 ± 0.08 | 97.99 ± 0.46 | 98.22 ± 0.09 | 96.39 ± 0.72 | 97.50 ± 0.06 |
| | MNIST-Add-Incomp | 93.48 ± 0.25 | 93.62 ± 0.82 | 94.18 ± 0.24 | 93.38 ± 0.39 | 93.54 ± 0.14 |
| | CUB | 98.98 ± 0.14 | 99.88 ± 0.04 | 99.82 ± 0.08 | 99.67 ± 0.10 | 98.87 ± 0.26 |
| | CUB-Incomp | 91.74 ± 0.19 | 95.37 ± 0.31 | 94.96 ± 0.26 | 93.22 ± 0.20 | 91.53 ± 0.29 |
| | CelebA | 56.99 ± 0.37 | 68.54 ± 0.66 | 66.48 ± 1.10 | 64.63 ± 0.93 | 44.37 ± 0.61 |
| 100% | MNIST-Add | 99.51 ± 0.04 | 99.51 ± 0.04 | 99.51 ± 0.04 | 99.51 ± 0.04 | 99.47 ± 0.06 |
| | MNIST-Add-Incomp | 94.99 ± 0.11 | 94.99 ± 0.11 | 94.99 ± 0.11 | 94.99 ± 0.12 | 95.03 ± 0.09 |
| | CUB | 99.90 ± 0.04 | 99.90 ± 0.04 | 99.90 ± 0.04 | 99.90 ± 0.01 | 99.85 ± 0.08 |
| | CUB-Incomp | 96.52 ± 0.19 | 96.52 ± 0.19 | 96.52 ± 0.19 | 96.56 ± 0.19 | 96.00 ± 0.21 |
| | CelebA | 70.02 ± 0.62 | 70.02 ± 0.62 | 70.02 ± 0.62 | 69.93 ± 0.72 | 53.39 ± 0.80 |

