# OpenReview forum: "Learning to Receive Help: Intervention-Aware Concept Embedding Models"
_NeurIPS.cc/2023/Conference — NeurIPS 2023 spotlight_

### Official Review · Reviewer_4KHx · 2023-06-16

**Soundness:** 2 fair
**Presentation:** 3 good
**Contribution:** 2 fair
**Rating:** 5
**Confidence:** 4

**Summary:**

The intervenability of concept bottleneck models has been taken for granted thus far. There have been numerous methods that tried intervening or propose policies to intervene. However, previous methods did not go so far as to optimize models for interventions. This work proposes intervention-aware training and extends Concept Embedding Models for this purpose. With experiments across MNIST, CUB, and CelebA, the authors demonstrate that IntCEM achieves on-par downstream performance compared to CEMs, and investigate the intervention performance with two sets of automated studies.

**After the rebuttal:** I read and acknowledged the authors rebuttal. I am leaning towards an acceptance position after the rebuttal, given that the authors clarified my questions. I find the paper's message interesting and the presentation to be clear and well-written, the reason I am not giving a higher score is the lack of real-world settings to test the method in more complex settings and where the models do not need concept annotations at the training time.

**Strengths:**

1. Optimizing a model for intervention awareness is an interesting and strong idea. Many works did indeed attempt interventions, whereas it is infrequent to optimize the model for this purpose and the intervenability has been taken for granted. I do have a question about whether the current form of this is useful (Q1), yet I still believe the idea is worth sharing with the community.

2. I believe having an intervention policy has implications beyond interventions per se. By looking at the Rollout Loss, I could imagine the policy revealing locally important / less important concepts and performing selective inference. While this is a hypothesis that needs verification, this could be an interesting direction to think about.

3. (Minor) I appreciate the clarity in the writing. Formalization is clear and easy to digest even though there are nuanced tricks to make the method work.

**Weaknesses:**

1. In my understanding, there is a disconnect between the promise/message of the paper and the experiments. Concretely;

- 1.1 In many parts of the paper, there is a theme around getting the models ready for expert feedback, which can make sense. However, I do not fully understand how we can infer that the current method achieves something related to a user or an expert.

- 1.2 In this regard, a major weakness is the lack of a user study. There are many arguments in the paper that touch upon optimizing the model for user interventions, whereas there are no experiments that verify this argument. An experiment where the users used these models, intervened on them, and IntCEM performed better than baselines would support the argument here.

- 1.3 Furthermore, the selection policy encodes a certain belief: That the users will choose concepts that would improve downstream performance. An alternative user metric would be showing that users prefer the concepts chosen by the selection policy. This may or not may not be the case. One could have also hypothesized that the policy can pick concepts that the users would also like to intervene on more than other policies, but again, there is no supporting evidence for this argument either.

In general, I believe this is an important disconnect that needs addressing.

2. The usability of the method suffers from the requirement to have concept annotations at training time, as do vanilla CBMs and CEMs. However, there are still concept-based models that could be worth exploring in this context that are already cited in the paper as [26, 27] which are Post-hoc CBMs and Label-Free CBMs that also scale to ImageNet/COCO. The authors make the argument that the method would complement those architectures as well, and I can believe this hypothesis, but this needs verification.

3. The above weakness also leads to another one, which is the lack of experimental settings closer to the real-world such as ImageNet, COCO, or medical datasets that were tested before in the above papers. I do think experiments would benefit from expanding beyond CUB and CelebA. Personally, I believe this would also allow impactful user studies.

4. It is hard for me to interpret the numbers in Sections 4.2 and 4.3. This is also connected to the 3rd point above – but the improvements in MNIST or CUB seem 1-2%. Further, as a minor comment, while there are Figures, all of the y-axes are in different scales, and I cannot really tell the numbers.

**Questions:**

1. In terms of the story, while the premise of intervention awareness sounds interesting at face value, I still think the case needs to be made for such a need. I believe it could help to give a concrete use case before laying out the method. E.g. L118-119 authors mention an interesting example, “may be hard for the model to predict on its own”. Can authors make it more concrete, give real world examples / formalize this? When would it hurt to not be intervention aware? When would we benefit from being intervention aware?

2. I would like to ask the authors’ reasoning about the first weakness I raised in the Weaknesses section. If intervention awareness is indeed to improve the interaction between the model and the human, how should we think about the current results? Is there a way to deduce that models are more ready for humans? Can we infer that humans would prefer this notion of intervention awareness over previous methods (or not having it)?

3. In my understanding, according to the loss function after L194, the selection policy $\psi$ is optimized to select the concept that would increase the probability of the ground-truth label being predicted the most, upon the intervention. Would this not bias the selection policy towards more typical interventions for the class (e.g. where $P(Y|C)$ is high)? For instance, say that a bird specie X is 95% of the time red but 5% of the time blue, whereas other birds are more often blue. Then, wouldn’t this loss function incentivize the selection policy to pick the color concept for blue X birds?

4. Similar to the above, while Section 4.3 studies the performance improvement from the policies, it is not clear what these policies pick. Do the authors have any analyses that explain what the chosen concepts are? For instance, are these the concepts that are specific to a class? Are these the most common concepts for the class, are there any other patterns there?

5. (Minor) I would appreciate having the Tables in addition to the figures, I’m sorry if I missed them but I could not find them in the Appendix either. Numbers could be hard to read from the Tables.

**Limitations:**

I think the most significant limitation is the lack of an actual real-world study to verify the claims. In my understanding, it is not possible to infer this from the current results. I would appreciate hearing the authors' reasoning about this.

---

> ### Author Rebuttal · Authors · 2023-08-08
>
> Dear Reviewer 4KHx,
>
> Thank you for your feedback and valuable questions. We appreciate that you found our work novel and clear.  Below we answer the questions you raised.
>
> ### Message of the paper and experiment design
>
> We believe there may be a misunderstanding of the methodology underlying our paper, particularly when seen in relation to prior work. IntCEMs are designed to better uptake user/expert feedback at test time (i.e., via “concept interventions”). However, we do not change how we think of concept interventions compared to prior work; intervention awareness is simply – yet powerfully – a training-time change that models that support concept interventions (in particular embedding-based models) can make use of to improve their receptiveness to interventions. We evaluate concept interventions in the same way as prior works in this area (e.g., [1-5]). The underlying assumption of our and their evaluation is that an expert can correctly answer a query of the form “What is the correct binary value of concept X?”; our evaluation is just a way to simulate such queries when we know the ground truth concepts. While there are important grounds for questioning these assumptions (see [5]) – in this first instantiation, we believe our experiments are substantial towards demonstrating the potential value of IntCEMs and follow the way the community has been evaluating concept interventions.
>
>
> ### Selection policy
>
> We emphasize that there is an important difference between what a policy may suggest and what the user may opt to intervene on. There is no assumption in our work dictating that a user must provide a concept label when requested by $\psi$. In fact, in Section 4.2 we show that our model significantly outperforms competing state-of-the-art methods even in the case where a user randomly selects which concepts to intervene on. Nevertheless, we agree that one may want to encode a user’s preferences into the intervention policy. This is an exciting direction and one we are keen to explore next with user studies (we will mention these two in a new future work section as suggested by reviewer 23Ef).
>
> We highlight, however, that in this work we operate under the reasonable and common assumption that expert queries can be asked and an expert will answer them correctly. This is no different to assumptions made in the active learning and active feature acquisition communities, both popular and mature fields of study.
>
>
> ### Intervention policy and concept typicality
>
> Notice that our learnt policy is dynamic (i.e., it is a function of the current sample and the previous interventions) rather than static (i.e., concept intervention is fixed a priori). As such, IntCEM’s policy $\psi$ does not pick the same sequence of concepts for all samples ( see Figure 1 in the rebuttal supplement for an example). Hence, it would not necessarily always pick the colour blue if the IntCEM is, say, very confident that the input sample is blue.
>
> ### Extension to other kinds of concept models and datasets
>
> As you correctly mentioned, the principles underlying IntCEMs could be applied to other models. We include details on possible extensions for traditional CBMs in our Appendix (see A.8) and discuss these results in our discussion. As for even more recent concept-based architectures such as post-hoc CBMs and label-free CBMs that are applicable to larger scale datasets lacking complete concept annotations (e.g., ImageNet and COCO), we learnt about them at ICLR 2023 only a couple of weeks before NeurIPS’ deadline. This near overlap with their publication and the NeurIPS paper submission deadline meant that we could not sensibly include them as part of our experiments and that these works fell within the two-month NeurIPS baseline exclusion period. Because of this, we focused on evaluating variants of CBMs and CEMs on datasets with a complete set of concept annotations across all samples. Nevertheless, we find these works very interesting and exciting, which is why we suggested as part of future work that one may explore applying IntCEM’s core design principles to these sorts of models. Further verification is sensible for future work, but we do not think this is essential for affirming the validity of our intervention-aware design. Finally, note that, unlike CBMs, IntCEM does not require a complete set of concept annotations because we its use of embeddings.
>
>
> ### IntCEM improvements and presentation of results
>
> We agree that the scales in our figures might make exact values hard to determine when analysing the gains obtained with IntCEMs. To address this, we have followed your suggestion and made a simplified table version of Figures 3 and 4 that we will include in a new appendix. Due to lack of space, we show only the table corresponding to Figure 3 (and only for some a fixed set of fractions of intervened concepts) in our attached supplement as Figure 1.
>
> Looking at this new table and Figure 3, we see a significant difference between IntCEMs and CEMs across all datasets once interventions are performed. For example, in CUB we observe a difference of 6.5% absolute improvement for IntCEM vs CEM when randomly intervening on 50% of the available concepts while in CelebA we observe a difference of ~13.20%. This table also shows that by intervening following IntCEM’s learnt policy $\psi$, IntCEM can achieve nearly perfect accuracy (99.17%) after intervening on 50% of the concepts.  This boost comes with negligible test time computational costs and without the need to (re)learn any further policies.
>
>
> ## References
>
> [1] Koh et al. "Concept bottleneck models." ICML 2020.
>
> [2] Kazhdan et al. "Now you see me (CME): concept-based model extraction." arXiv:2010.13233 (2020).
>
> [3] Espinosa Zarlenga et al. "Concept embedding models" NeurIPS 2022.
>
> [4] Chauhan et al. "Interactive concept bottleneck models." AAAI 2023.
>
> [5] Collins et al. "Human Uncertainty in Concept-Based AI Systems." arXiv e-prints (2023).

---

> > ### Comment · Reviewer_4KHx · 2023-08-10
> > **Response to the Rebuttal**
> >
> > I thank the authors for their detailed rebuttal.
> >
> > > Message of the paper and experiment design
> >
> > Thank you for the discussion. I do believe now I better understand the notion of intervention here.
> >
> > > Policy
> >
> > Thank you for the clarifications. Of course, I see that the policy is dynamic and not a static one. My typicality question was a very simple and empirical one - in practice what concepts end up being intervened on, and are they simply the typical concepts? This should be a simple statistic compute, if I am not mistaken. If the answer is yes, I think this has important implications.
> >
> > > Extension to other kinds of concept models and datasets
> >
> > Thank you for this discussion. I find the argument authors raised to be completely fair. Due to the overlap in the exclusion period and moving forward I will not consider the inclusion of P-CBM or Label-Free CBM results. I would still find it informative to add a (short) discussion of this in the limitations. Similarly, I still find the lack of larger datasets to be a limiting factor to demonstrate how much the method would transfer to real world problems.
> >
> > > IntCEM improvements and presentation of results
> >
> > Thank you for creating this table.
> >
> >
> > > Overall Comment
> >
> > I thank the authors for an informative rebuttal. I find several of the authors' arguments convincing and will revise my score accordingly. I would still appreciate if it were possible for the authors to answer the simple question I asked above.
> >
> > To clarify, while I am leaning towards an accept position and find the ideas interesting, the reason I do not increase my score further is due to the lack of real-world settings (larger datasets, more complex problems that would clarify that results transfer beyond MNIST/CUB) and user studies which would strengthen the significance of the work.

---

> > > ### Author Response · Authors · 2023-08-12
> > >
> > > Thank you so much for your reply and for leaning towards acceptance after our rebuttal. We really appreciate the time taken to reply to our rebuttal and your careful answer. Below we reply to your specific concerns. Please let us know if there are any further concerns and/or questions after our replies below.
> > >
> > > ### Typicality
> > >
> > > Apologies on our end for misunderstanding your concern. We now see what you mean, and we agree with you. Empirically, we observe that certain concepts are selected with a higher probability at earlier steps of rollouts of our policy $\psi$. For example, we observed that "breast pattern" and "wing shape" were two concepts that were often selected in the first few steps by $\psi$ for the IntCEM we used to generate the plots in Figure 1. Intuitively, these seem to be highly informative concepts for some underrepresented classes. This is indeed an interesting observation, and we will discuss it in Section 4.3 while also including a histogram showing these statistics as part of the appendix in our updated manuscript. Nevertheless, for the sake of fairness, we are not allowed to include this histogram as part of this reply, and therefore we apologize that we cannot provide this figure at this moment.
> > >
> > > ### Large datasets and user studies
> > >
> > > Our evaluation consisting of studies on task performance, concept performance, receptiveness to different intervention policies, and study of our own learnt intervention policy across five tasks (three of which are real-world datasets) unfortunately left us with very little room for adding even more baselines, datasets, or well-crafted user studies. Nevertheless, we hope that the publication of our method in an easy-to-use Pytorch-based installable library will enable future research to be easy to carry regardless of the task or setup. As you suggest, however, we will make sure to include a short discussion of the need for future larger-scale studies, as well as user studies, in our limitations subsection.

---

### Official Review · Reviewer_23Ef · 2023-07-06

**Soundness:** 4 excellent
**Presentation:** 4 excellent
**Contribution:** 4 excellent
**Rating:** 9
**Confidence:** 5

**Summary:**

The authors proposed a modified version of CEM that is better at receiving human test-time intervention by explicitly incorporating intervention into the training stage. Specifically, an intervention prediction module is trained to behavior clone an optimal-greedy intervention policy (Skyline). The CEM is trained to optimize for the correctness of the initial guess as well as the prediction after the predicted intervention. The idea of the method is simple and straightforward. Extensive experiments explore design choices and show strong results in practical settings.

**Strengths:**

* The paper is well-written. Extremely easy read and all concerns I have about the method are covered in experiments.
* Simple and effective idea anchored on the premise that models not explicitly trained to receive intervention might not handle intervention well (empirical support in  Fig 3, 4).
* Questions proposed in P6L221 - 230 effectively guides reader through the work and the authors' through process.
* The authors care about how this method could realistically be applied in real-world settings.
    * Heuristics for tuning hyperparameters for the losses are provided.
    * Considers realistic settings of applying IntCEM where human-intervention might not be greedy-optimal and show empirically that IntCEM still performs better than baselines under non-optimal interventions (even random intervention).
    * Observe desirable properties for application in the real world (P7L265)

**Weaknesses:**

* The only weakness is lacking human study but this could be conducted rather straightforwardly. Given the extensive study on intervention policies, I believe this to be merely nitpicking.
* This is obviously a work that would induce plenty of follow-up work. Perhaps adding a section for what types of design choices have not been explored yet (e.g. other RL methods besides BC) could help stimulate the research community.

**Questions:**

* How severe is information leakage for CEMs? This is tangent to this work's contribution but since this work is based on the CEM framework, perhaps the authors could comment on whether IntCEM succeeds because of the leakage of downstream task information into the concept embeddings?
* Typo:
    * P2L39 artefact -> artifact

**Limitations:**

Yes.

---

> ### Author Rebuttal · Authors · 2023-08-08
>
> Dear Reviewer 23Ef,
>
> Thank you so much for the very encouraging review and the valuable feedback and suggestions. We are glad you found our work very easy to read, well-motivated, and friendly for reproduction/real-world use. Below we discuss some of the points you raised in your review, including answers to your questions (which led to what we believe are some important new additions to our manuscript!).
>
> ### Section detailing future work
>
> We agree this is a useful addition. We will include this section in our updated manuscript outlining some of the future directions we have discussed in our rebuttal with you (e.g., user studies and a better understanding of leakage) and the other reviewers (e.g., evaluation of large-scale concept-based models).
>
> ### Lack of user study
>
> Please see our global rebuttal reply for a reply to this concern.
>
> ### Information leakage and IntCEMs
>
> We believe that your intuition on how leakage may play a part in IntCEM is correct and we present some evidence in favour of it in our global reply for this rebuttal. These results will be included as a new appendix in our updated manuscript.

---

> > ### Comment · Reviewer_23Ef · 2023-08-17
> >
> > Many thanks to the authors for the response. I do agree that information leakage may or may not be (un)desirable and explicitly stating this in the paper is a good way to inform readers about the potential catch with this method working.

---

### Official Review · Reviewer_oMUi · 2023-07-08

**Soundness:** 4 excellent
**Presentation:** 4 excellent
**Contribution:** 4 excellent
**Rating:** 9
**Confidence:** 4

**Summary:**

The authors in this paper proposed a novel method of improving test-time interventions for Concept Bottleneck Models (CBMs). Although many CBM works showcase their ability to do intervention, none of them explicitly motivate the learned model to do well on intervention during training phase, hence hindering their ability and reliability to do intervention correctly. This work builds on top of concept embedding models (CEMs) and constructs an additional probability distribution (parameterized by neural networks) to learn what concepts to intervene on. To learn this distribution, the authors proposed a composite of three losses, where the first loss (Rollout loss) follows a greedy strategy of selecting concepts that yield the maximum probability of the given class, the second (task prediction loss) penalizes wrong predictions with many interventions, in which the penalty scales with the number of intervention performed. Finally, the third loss concept loss is just binary cross entropy to promote the correctness of the concept prediction. Empirical results on MNIST and CUB (and their variants) show that the proposed formulation performs competitively against state-of-the-art CEMs/CBMs that also utilizes interventions to improve performance. The authors also argued that the sampling from the learned distribution is much more scalable than current state-of-the-art methods.

**Strengths:**

The paper is well-written and is organized in a chronological and clear manner. Introduction sufficiently motivates the problem formulation. The background and related work is written neatly so that the paper is sufficiently self-contained. The description of the method and notation is also clear and easy to follow.  And experiments have clear objectives and empirical results clearly demonstrate the effectiveness of their method.

Given CBMs are a very popular topic, and that there is a lack of work in the current field that focuses on improving test-time interventions in CBMs, I consider the formulation proposed in this work  of significantly novel. It definitively addresses a limitation of CBMs/CEMs and proposed a good formulation that addresses that limitation.


**Weaknesses:**

There are no major weaknesses in this paper. But a few things that I suggest the authors to add for the sake of completeness:

As the main selling point of interpretable-by-design methods are interpretability, it feels a bit odd that authors did not add any examples of interventions on a particular sample. Even though the metrics clearly show the method is effective, a few concrete test samples and use of intervention can better illustrate what interventions do, how the posterior of the task prediction changes before/after interventions, etc. Potentially, illustrations could help readers who may perhaps be from a different field.


**Questions:**

1. I understand that “the single MC sampling per training step” is valid, but what is the main motivation for doing so? Is it to improve training speed? And do you speculate, (even better, show with ablation studies) that improving this estimation can improve convergence when optimizing your objective?
2. In Figure 3, can you explain  why intervention could cause a drop in performance for Join CBM-Logit? Since the proposed objective explicitly optimizes for CE after intervention, would you agree that your method trained to prevent a drop in performance after intervening from happening?
3. How long does it take to train typical model with the proposed objective? I would imagine the rollout loss can be slow when T is large.  Does restricting T to be small heavily impacts the performance of the trained model?


**Limitations:**

The limitations of this work is addressed in the main manuscript and is well-stated. The answer to my questions above can potentially brought up as limitation, and if so, I would suggest the authors to add that to the limitation section.

---

> ### Author Rebuttal · Authors · 2023-08-08
>
> Dear Reviewer oMUi,
>
> Thank you so much for the very encouraging review and the extremely valuable feedback that came with it. We are very glad that you found our work significantly novel, well-written, and clearly motivated. Below we answer the questions you raised in your review.
>
> ### Motivation for single MC sample
>
> As you rightly suggested, the primary motivation for a single MC sample is efficiency at training time (as this has to be done for each training step). During early experimentation with our method, we did some informal studies on how the number of samples affected our model’s performance. We noticed that the gains from more MC samples are marginal in practice yet have a significant cost to wall-clock training times. Intuitively, we believe this can be because a sample is seen multiple times during training (across different epochs) and therefore there is not much benefit in adding more mask samples during a specific epoch if the same sample will be eventually seen again in a future epoch (and we will sample different initial intervention mask then). One way to think of this effect is akin to how increasing MC samples during a VAE’s training has marginal improvements while making training times longer. As such, we opted for a single MC sample.
>
> ### Detrimental interventions in Joint-CBM-Logit
>
> This is an excellent and interesting question. Our hypothesis is that traditional CBMs can react negatively after an intervention is performed because of *concept leakage* (see our global reply for more information regarding this). Leakage occurs when a CBM-like model learns to exploit a concept’s continuous nature to encode unnecessary information as part of a concept’s representation. Such impurities can have detrimental effects when intervening especially if the intervention operation results in a destructive state of this continuous space (e.g., setting a concept probability to $1$ or $0$ as done when intervening in a Joint-CBM). This is because such an operation would destroy any extra information that the downstream label predictor might’ve otherwise exploited to predict the output class, leading to detrimental concept interventions. When using logits over sigmoidal bottlenecks, we allow much more flexibility in the concept activations and therefore the chance for concept leakage is higher (and also a possible reason why the same drop in Figure 3 is not observed for Joint-CBM-Sigmoid). Notice furthermore that when the set of concept annotations at training time is incomplete w.r.t. the downstream task (as in CUB-Incomp), the chances of leakage in traditional CBMs are much higher as the model’s bottleneck will be underprovided and therefore the model will have to make trade-offs between task accuracy and concept purity/accuracy.
>
> Such leakage occurs in IntCEMs; however, as you mention, it is unlikely to have a detrimental effect on the model’s performance after the intervention is performed due to how IntCEM is trained and how our model operates when intervened on. This is because (1) we incentivise our model to be reactive to interventions at train time with the hope that they will learn to be more accurate with very little external feedback at test time, and, more importantly, (2) interventions in IntCEMs are **not destructive** as we only change the embedding that we pass into the downstream label predictor, allowing the model to still exploit extra information encoded in the learnt embeddings even after the intervention has been performed.
>
> We realise this could have been better explained in our experimentation section and will include this leakage-motivated discussion and hypothesis in our updated discussion section.
>
> ### Training time of IntCEM and effect of $T$
>
> We studied the effect of IntCEM’s regularisers on its training times in Appendix A.5 (particularly Table A.2). To summarise, we observed that the training times in IntCEMs increase by about 60% compared to CEMs. However, it is worth noting that we observed a large variance in training times (with even some IntCEM runs converging faster than CEM runs!) and that the extra training costs of IntCEMs amortise in practice as running the learnt intervention policy is much more efficient than equivalent post-hoc intervention policies (see table A.3 for details). Training time computations are also implementation-specific so they could be further optimised in the future.
>
> As for the value of $T$, you are correct in stipulating that this affects both training times and performance. We included an ablation study showing the effect of $T$ in our model’s performance in Appendix A.6.3 (particularly Figure A.3). There, we observe significant performance differences only when T is very small (e.g., $T=1$) relative to larger values of $T$ (although notably, IntCEM is still better than CEM in this instance after a larger number of interventions). In practice, we observe that increasing $T$ above a small value (around $5$) results in negligible gains. Some intuition for this result is that one has fewer “interesting” concepts for a given sample to select from after long trajectories, leading to diminishing returns as we increase the value of $T$.

---

> > ### Comment · Reviewer_oMUi · 2023-08-12
> > **Thank you**
> >
> > Thank you for addressing my questions. As of now, I do not have any further questions.

---

### Official Review · Reviewer_Zgwp · 2023-07-08

**Soundness:** 3 good
**Presentation:** 3 good
**Contribution:** 2 fair
**Rating:** 7
**Confidence:** 4

**Summary:**

The authors introduce IntCEMs, an extension of Concept Embedding Models designed
specifically to react correctly to external interventions to the learned concepts.
Compared to regular CEMs, IntCEM feature two additional elements:  a policy that,
essentially, guesses what interventions a human expert would do on the model for
any given input, and a penalty that regularizes the model to achieve high accuracy
under interventions.  The idea is to ensure the model performs well even after
interventions - under the assumption the model requires interventions to ``ask
for help'', i.e., to obtain the ground-truth label of certain concepts at test
time.  Experiments show that in fact IntCEMs outperform CEMs and CBMs on four
data sets at test time under interventions, even from a different policy.

**Post-rebuttal update**: Increasing the score by one point under the assumption
the authors will fairly highlightly leakage as a limitation of CBMs/CEMs/IntCEMs
and its interaction with interventions in the paper.  I am still a bit concerned about
impact, but the contents of the paper are good quality overall.

**Strengths:**

+ English is good, narrative is generally good but parts of it are confusing, see below
+ Motivation is sensible
+ Proposed approach is also sensible
+ Empirical evaluation is generally positive, with only some minor drops in concept quality depending on choice of \lambda_roll
+ Related work is sufficient

**Weaknesses:**

- One issue is limited significance, in the sense that the authors tackle a problem that affects only a certain type of operation (namely, a *specific* type of interventions - not all possible interventions) performed on a certain type of model (CEMs, CBMs).  It is true that the performance gain of IntCEM vs CEM under interventions can be substantial (CelebA in figure 3), but it is generally more modest (CUB, CUB-Incomp; the diff in perf for MNIST-Add and MNIST-Add-Incomp are somewhat biased by the difference at Groups Intervened = 0, which is not due to extra robustness to interventions).  This is not a deal breaker, and indeed my score is positive, but it explains why I decided not to go above weak accept.

- Writing is a bit confusing.  It took me a while to understand where the ``learning to receive help'' fits into the abstract and introduction.  The idea is that models can ask users to help them at test time by - essentially - requesting the ground-truth label of certain concepts, which means that this work is closer in spirit to concept-level active learning (but at test time) rather than to interventions proper.  These have a causal connotation and can be used for changing the concepts to *any* value, not only to their ground-truth values.  See for instance all the literature on algorithmic recourse.  The target task should be made more explicit, at the bare minimum in the introduction.  I also feel the text abuses greek letters (all of section 3) a little, hindering readability.  Every time I see, say, \eta, \psi, \omega... in the text, I have to look back at their definition.  The $c$ letter also appears in too many variants.  More generally, the text feels quite compressed.

- Evaluation is only carried out against CEMs and CBMs.  Other self-interpretable models exist and have been published recently.  Considering the focus on CBMs, this is not a major issue, but it is an issue nonetheless.

Minor issues worth fixing
-------------------------

- line 102: the range {0, 1}^k is wrong, \tilde c_i is set to 0.5.

- Considering \mu_i also appears in the definition of \tilde c_i, Eq 1 is probably too complex for what it needs to achieve.  Is it possible to   simplify it?

**Questions:**

Q1. It was shown by:

  Mahinpei et al. "Promises and pitfalls of black-box concept learning models", 2021.

  Margeloiu et al. "Do Concept Bottleneck Models Learn as Intended?", 2021.

that CBMs suffer from "concept leakage", whereby learned concepts encode information from unintended sources (like unobserved concepts), with negative consequences of interpretability.  I am wondering whether 1) How IntCEMs fare in this regard; specifically, I am only aware of two self-interpretable models that specifically address this, namely:

  Havasi et al. "Addressing Leakage in Concept Bottleneck Models", 2022.

  Marconato et al. "GlanceNets: Interpretable, Leak-proof Concept-based Models", 2022.

A brief comment on IntCEMs vs leakage would be welcome - this is an important issue.  2) Robustness to interventions can prevent leakage - the link is that the model is specifically trained to be robust against replacing a (possibly leaky) concept prediction with a (presumably non-leaky) concept annotation.

Q2. Why did you compare only against CEMs and CBMs?  There are other self-interpretable models out there, like the two aforementioned works, part-prototype networks, concept whitening, self-explainable neural networks, and many others.  They all support interventions.

**Limitations:**

I think so, in section 5 there is a brief discussion on some limitations (e.g. time complexity).  The issue of concept leakage is not mentioned though.

---

> ### Author Rebuttal · Authors · 2023-08-08
>
> Dear Reviewer Zgwp,
>
> Thank you for your very insightful feedback. We are glad you found our paper’s motivation, evaluation, and relevant work interesting and generally positive. Below we address your questions.
>
> ### More explicit definition of  “interventions”
>
> This is a great point. We want to clarify that, as you correctly mentioned, the term “intervention” here is not used as in the causality literature and is more related to the field of active feature acquisition (e.g. [1]). This aligns with your intuition regarding seeing this as a form of concept-level active learning at test time. Nevertheless, the concept learning community widely uses the term “concept intervention” for the operation in which an expert provides a binary concept label at test time for a CBM-like model. Hence we maintained the term to avoid confusion within this paper’s “host” field of study.
>
> To address possibly confusing future readers from other subfields, we will state what we mean with the term “concept interventions” early on in our introduction. Furthermore, we will elaborate on how these differ from their causal counterparts and how they are related to active feature acquisition in our background section.
>
> ### IntCEM’s performance gains
>
> The scale in some tasks in Figures 3 and 4 is large as we needed to fit baselines underperforming w.r.t. IntCEM. However, looking at CUB’s performance values in a table view of these figures (see Table 1 of our supplement), we observe a significant absolute boost of 6.5% for IntCEM vs CEM when randomly intervening on 50% of concepts (we achieve an almost perfect performance after randomly intervening on 75% of the concepts). What is more important, however, is that by intervening following $\psi$ rather than performing just random interventions, IntCEM’s absolute gains in CUB when intervening with 50% of the concepts shift to ~9.5% (with an almost perfect acc of  99.17%).  This boost comes with negligible test time computational costs and without the need to (re)learn any further policies.
>
> As for the MNIST tasks, these tasks showcase how our training procedure can lead to richer embeddings that allow **performance boosts even without interventions**. We will add a small sentence to highlight this in Section 4.1. Moreover, to make this analysis easier for external readers, we will include Table 1 of our rebuttal supplement in a new Appendix.
>
> ### Evaluation baselines
>
> To the best of our knowledge, the main non-graph-based prior works that have evaluated concept interventions are CBMs (they introduced concept interventions formally), Concept-based Model Extraction (CME) [2], CEMs [3], and leakage-free CBMs [4]. Leakage-free CBMs avoid using soft representations that are prone to leakage by using hard representations for their bottleneck together with an optional side channel and training procedure that intentionally “flows/leaks” information not captured by the hard known concepts. Nevertheless, although these additions are noteworthy and interesting, this model effectively operates like a CBM in practice and the extra leakage bypass it offers is a constrained version of the bypass allowed by embeddings in CEMs. Similarly, CMEs were shown to underperform vanilla CBMs when intervened on [2] and are post-hoc explainability methods rather than interpretable architectures. Because of these reasons, we focused on CEMs and CBMs as they capture the key mechanisms in all of the methods mentioned above.
>
> We indicate why we did not include other concept-based interpretable methods as baselines for our evaluation:
>
> - GlanceNets: Although GlanceNets support concept interventions, these go beyond the original scope of the architectural changes between GlanceNets and CBMs. The lack of intervention experiments in GlanceNet’s paper supports this. Furthermore, concept interventions in GlanceNets are not too different from those in Hybrid-CBMs [3], which are known to underperform w.r.t CEMs.
>
> - Concept Whitening (CW): To the best of our knowledge, Concept Whitening does not support concept interventions as it is unclear how one would set entire feature maps at test-time to trigger a concept intervention. This difficulty is even greater when one considers that CW usually performs a reduction over these maps to determine a concept’s “activation level”. Notice that their manuscript does not allude to concept interventions as they slightly precede CBMs.
>
> - Self-explaining Neural Networks (SENNs) and Prototypical part networks (ProtoPNets): Both of these methods work in a concept unsupervised manner (they do receive feedback from task labels but not from concept labels). Therefore it is not immediately clear how to perform concept interventions given that one needs to assign semantics to the discovered concepts first. Such a task is non-trivial to evaluate and goes beyond the scope of our own work.
>
> - Other more recent methods (label-free CBMs, post-hoc CBMs): There are very recent methods that do support concept interventions (e.g., label-free CBMs and post-hoc CBMs). However, these were published less than two months before the submission deadline (at ICLR 2023), which meant that we could not reasonably include them as part of our evaluation.
>
> We will clarify these choices in our evaluation section by justifying our baselines as done above.
>
> ### Misc minor fixes
>
> Thanks for finding this typo! As for Eq 1, it could be simplified by using a piecewise definition, however our original intent of expressing Eq 1 as it is was to make it clear how one may right a concept intervention operation in a differentiable manner.
>
> ## References
>
> [1] Li et al. "Active feature acquisition with generative surrogate models." ICML 2021.
>
> [2] Kazhdan et al. "Now you see me (CME): concept-based model extraction." arXiv:2010.13233 (2020).
>
> [3] Espinosa Zarlenga et al. "Concept embedding models" NeurIPS 2022.
>
> [4] Havasi et al. "Addressing leakage in concept bottleneck models." NeurIPS 2022.

---

> > ### Comment · Reviewer_Zgwp · 2023-08-12
> > **Reply**
> >
> > > More explicit definition of “interventions”
> >
> > I do appreciate this change, thank you.
> >
> > > IntCEM’s performance gains
> >
> > Thank you for the clarification.
> >
> > > Leakage-free CBMs [...] effectively operates like a CBM in practice and the extra leakage bypass it offers is a constrained version of the bypass allowed by embeddings in CEMs.
> >
> > Based on your general reply, would I be correct in thinking that a "less leaky" model is likely to see smaller benefit from better chosen interventions? I'm just curious.
> >
> > > concept interventions in GlanceNets are not too different from those in Hybrid-CBMs [3], [...]
> >
> > My understanding is that Hybrid-CBMs are CBMs with additional unsupervised concepts. Presumably these are used for prediction. GlanceNets do not use unsupervised concepts for prediction. In what sense are their "concept interventions" similar?  Also, what sense Hybrid-CBMs "underperform"?
> >
> > > Concept Whitening, SENNs, ProtoPNets
> >
> > I see.  I think these points are valid, at least at a high level.  Thank you for the explanation.
> >
> > Edit: fixed the formatting.

---

> > > ### Author Response · Authors · 2023-08-12
> > >
> > > Thank you for replying to our rebuttal; we really appreciate your feedback and the thorough review. Below we reply to the questions raised in the comment above:
> > >
> > > >  "Would I be correct in thinking that a 'less leaky' model is likely to see smaller benefit from better chosen interventions?"
> > >
> > > This is a really good question. We think a thorough study of this may be needed to answer your question. However, if we were to speculate, our intuition from this work is that models that allow more information through their bottlenecks, and that are able to maintain that information after an intervention is performed (i.e., the intervention is not destructive), may be able to perform better after interventions. This is because the label predictor has more information it is able to exploit after an intervention is performed.
> > >
> > > Nevertheless, we emphasize this is **not** a sufficient condition as seen in Hybrid-CBMs. In these models, as one increases the amount of extra unsupervised capacity in the bottleneck, interventions become less effective due to (1) the model learning possible shortcuts directly from the input features to the task prediction via the unsupervised neurons, and (2) the lack of clarity on how one should modify these unsupervised neurons when one intervenes on a specific known concept. Notice that CEMs and IntCEMs circumvent this shortcutting by introducing training-time regularisers that implicitly encourage the distribution of concept embeddings to be distinct when the concept is "on" than when it is "off".
> > >
> > > >  Hybrid-CBMs and GlanceNets
> > >
> > > You raised a good point, and we acknowledge that GlanceNets do not use the extra unsupervised "concepts" for prediction, while Hybrid-CBMs use the unsupervised capacity in their predictive process. We meant that they were similar in that, during interventions, only the supervised concepts are modified (without touching the unsupervised neurons in the bottleneck). This leads to "ill-defined" states during interventions as the change intended from intervention is not propagated to all relevant neurons in the bottleneck. The original CEM paper empirically showed that this leads to Hybrid-CBMs being significantly less receptive to interventions than all other baselines they compared against (including CBMs and CEMs). If GlanceNets would use the unsupervised concepts during prediction, we would expect a similar situation to develop. More importantly, however, precisely because GlanceNets do not use the unsupervised concepts for prediction and they use them only for reconstruction for the open set recognition, we expect interventions to suffer significantly when the set of training concepts is incomplete w.r.t. the downstream task, a common setup in real-world tasks (e.g., $\texttt{MNIST-Add-Incomp}$, $\texttt{Cub-Incomp}$, $\texttt{CelebA}$).  Because of this, and equally as important because GlanceNets were not designed or even evaluated to intake interventions, we decided to focus our evaluation on models that are very likely better at receiving interventions such as CEMs.

---

> > > > ### Comment · Reviewer_Zgwp · 2023-08-14
> > > > **Reply**
> > > >
> > > > Thank you again for engaging in the discussion.  I'll keep your points in mind when discussing with the other reviewers.  To be clear, I still think the problem being tackled is relatively minor, however the paper's contents are definitely there.  Given the authors promised to discuss leakage in more detail, and assuming they will discuss this as a limitation of IntCEMts, I will increase my score to 7.

---

### Author Rebuttal · Authors · 2023-08-08

We thank the reviewers for their very insightful feedback and for taking the time to read our work carefully. Their feedback has certainly improved the quality of our manuscript. We hope to address your concerns in this rebuttal and its corresponding supplementary document. We reply to questions shared by two or more reviewers in this general rebuttal while replying to specific questions raised by reviewers in their respective rebuttals.

# Summary of Supplement

In the rebuttal’s supplement (see PDF attached below), we include the following figures and tables:

- Table 1 represents a (simplified) tabular version of Figure 3 to help better analyse each model’s performance when a fixed fraction of concepts are intervened on. This table will be added to the Appendix in our updated manuscript.

- Figure 1 depicts an application of IntCEM’s learnt policy to two CUB test samples. This figure will be added in the main body of our updated manuscript in experimental Section 4.3 where we discuss IntCEM’s learnt policy $\psi$.

- Table 2 shows the Oracle Impurity Score (a measurement of leakage described below) for our jointly-trained baselines across all datasets. This constitutes a **new experiment** introduced in this rebuttal, the purpose of which is to understand better how leakage works in IntCEM and the role it may play in concept interventions with embedding-based models. We will include this table in a new Appendix section with a brief summary of its main results included in our discussion section.

# Answers to common questions

### Concrete example of IntCEM’s policy at test-time (reviewers oMUi and 4KHx)

In Figure 1 of the supplement, we show two concrete examples of intervening on an IntCEM in CUB to help clarify how our model works and how following its policy $\psi$  can change its posterior label distribution in practice. These examples will be included in Section 4.3  where we discuss how $\psi$ performs.

### Leakage in IntCEM’s embeddings (reviewers Zgwp and 23Ef)

As mentioned by reviewers Zgwp and 23Ef (and also related to one of reviewer oMUi’s questions), leakage [1] is a known issue in traditional CBMs. In fact, it has been empirically shown by [2] that leakage may lead to detrimental interventions because a CBM’s downstream label predictor may learn to rely on information leaked from concept representations and this information is necessarily destroyed when performing interventions in CBMs. This is because to intervene in a CBM’s bottleneck, one must overwrite a neuron’s activation to one that is aligned with the ground truth value of the concept that is represented by that neuron (e.g., a bottleneck’s neuron may be set to $0$ if the concept it represents is known to be “off” for the current sample, removing all information that could have been encoded in the continuous value previously outputted by that neuron).

 This issue does not arise in CEMs and IntCEMs because **embedding-based concept interventions are not destructive**. Indeed, as described in Section 2 of our paper, when intervening on either of these models, one essentially performs a swap between embeddings, allowing the model to maintain any leaked information from the input/other concepts/task labels as part of the embedding of each concept. Therefore, we do not expect interventions to be significantly affected by leakage in concept representations as long as a concept’s distribution of positive ($\hat{\mathbf{c}}^{(+)}$) and negative ($\hat{\mathbf{c}}^{(-)}$) embeddings are easy to discriminate by the downstream label predictor.

As keenly pointed out by reviewer 23Ef, we hypothesize that higher leakage may contribute to IntCEMs’ better performance when being intervened on. To evaluate this hypothesis, we measure the Oracle Impurity Score (OIS) [2] of concept representations learnt by CBMs, CEMs, and IntCEMs across all tasks. This score, between 0 and 1, essentially measures how much extra information, on average, each learnt concept representation captures from other possibly unrelated concepts (with higher scores representing higher impurities and therefore more leakage between concepts). Our results, shown in Table 2 of our rebuttal’s supplement, show that (1) as one would expect, there is significantly more leakage in CEMs compared to CBMs and, (2) more importantly, IntCEM embeddings are also capturing more impurities than CEM’s embeddings across all datasets we tested except for $\texttt{CUB-Incomp}$, providing evidence towards our hypothesis. Given that this preliminary leakage study may open the door for potentially useful and interesting future research, we will include this experiment in a new appendix (with a brief discussion of it introduced in our paper’s discussion). Our updated discussion will highlight that contrary to common assumptions, leakage may not always be undesired and could be a healthy byproduct of more expressive concept representations in models that accommodate such expressivity.

### User studies  (reviewers 23Ef and 4KHx)

We agree that a human user study could reveal very interesting results and complement our paper's extensive experiments and main contributions.  We have, however, left this for future work as we believe some exciting research directions spawning from a formal user study for concept interventions would merit a whole paper on their own. We are indeed very excited and optimistic about exploring these future avenues.

Following Reviewer 23Ef’s suggestion of including a future work section for our manuscript, we will update our paper to suggest user studies as part of future work.

### References

[1] Mahinpei et al. "Promises and pitfalls of black-box concept learning models." Workshop on Theoretic Foundation, Criticism, and Application Trend of Explainable AI at ICML (2021).

[2] Espinosa Zarlenga  et al. "Towards robust metrics for concept representation evaluation." AAAI (2023).

---

> ### Comment · Reviewer_Zgwp · 2023-08-12
>
> Thank you for the detailed reply.  I have only one comment.
>
> > [...] leakage may lead to detrimental interventions because a CBM’s downstream label predictor may learn to rely on information leaked from concept representations [...]
> > This issue does not arise in CEMs and IntCEMs because [...] one essentially performs a swap between embeddings, allowing the model to maintain any leaked information from the input/other concepts/task labels as part of the embedding of each concept.
>
> This is a fair point.  However, based on your reply, I take that IntCEMs are susceptible to concept leakage.  This is an issue in and of itself.  Of course, leakage is not the focus of this paper and I do not expect the authors to drastically change the method or the text to address it.  But I think it is worth informing readers of this issue (as it /does/ impact interpretability of the learned concepts) and that there exist works that specifically address it.  In other words, I would be disappointed if the authors chose to dismiss it.

---

> > ### Author Response · Authors · 2023-08-12
> >
> > Dear Reviewer Zgwp,
> >
> > Thank you for getting back to us regarding this point. We agree that leakage may have implications beyond allowing IntCEMs/CEMs to have higher accuracy, and those should be properly studied in what could probably be a paper on its own. As discussed in our rebuttal, we will discuss our findings regarding leakage in our updated discussion section and include the experiment mentioned above in a new appendix. Nevertheless, our paper's main claims, results, and observations still hold. Moreover, we believe this preliminary leakage study on concept-based architectures, together with the discussion we will include, will open up the door for immediate future research and increase the potential impact of this work. For that, we thank you and the other reviewers for bringing this to our attention.

---

> ### Author Response · Authors · 2023-08-21
> **Thank you**
>
> We would like to thank all reviewers for their valuable reviews and for engaging with us in the discussions that followed. Your feedback has allowed us to significantly improve our manuscript.

---

### Decision · Program_Chairs · 2023-09-21

**Decision:**

Accept (spotlight)

**Comment:**

This is a very well-written paper, with a significant novel contribution. The method is well explained and the experiments are well designed and demonstrate the effectiveness of the proposed model. One limitation is the the lack of an actual real-world study to verify the claims, which would be nice to see in follow-up work.